



# Interhemispheric differences of mesosphere/lower thermosphere winds and tides investigated from three whole atmosphere models and meteor radar observations

Gunter Stober[1], Ales Kuchar[2], Dimitry Pokhotelov[3], Huixin Liu[4], Han-Li Liu[5], Hauke Schmidt[6], Christoph Jacobi[2], Kathrin Baumgarten[7], Peter Brown[8,9], Diego Janches[10], Damian Murphy[11], Alexander Kozlovsky[12], Mark Lester[13], Evgenia Belova[14], and Johan Kero[14]

[1]Institute of Applied Physics & Oeschger Center for Climate Change Research, Microwave Physics, University of Bern, Switzerland
[2]Institute for Meteorology, Universität Leipzig, Leipzig, Germany
[3]Institute for Solar-Terrestrial Physics, German Aerospace Center (DLR), Neustrelitz, Germany
[4]Department of Earth and Planetary Science, Kyushu University, Japan
[5]High Altitude Observatory, National Center for Atmospheric Research, Boulder, CO, USA
[6]Max Planck Institute for Meteorology, Hamburg, Germany
[7]Fraunhofer Institute for Computer Graphics Research IGD, Rostock, Germany
[8]Dept. of Physics and Astronomy, University of Western Ontario, London, Ontario, Canada N6A 3K7
[9]Western Institute for Earth and Space Exploration, University of Western Ontario, London, Ontario, N6A 5B7, Canada
[10]ITM Physics Laboratory, Mail Code 675, NASA Goddard Space Flight Center, Greenbelt, MD 20771, USA
[11]Australian Antarctic Division, Kingston, Tasmania, Australia
[12]Sodankyla Geophysical Observatory, University of Oulu, Finland
[13]University of Leicester, Leicester, UK
[14]Swedish Institute of Space Physics, Kiruna, Sweden

**Correspondence:** gunter.stober@iap.unibe.ch

**Abstract.** Long-term and continuous observations of mesospheric/lower thermospheric winds are rare, but they are important to investigate climatological changes at these altitudes on time scales of several years, covering a solar cycle and longer. Such long time series are a natural heritage of the mesosphere/lower thermosphere climate, and they are valuable to compare climate models or long term runs of general circulation models (GCMs). Here we present a climatological comparison of wind observa-

5 tions from six meteor radars at two conjugate latitudes to validate the corresponding mean winds and atmospheric diurnal and semidiurnal tides from three GCMs, namely Ground-to-Topside Model of Atmosphere and Ionosphere for Aeronomy (GAIA), Whole Atmosphere Community Climate Model Extension (Specified Dynamics) (WACCM-X(SD)) and Upper Atmosphere ICOsahedral Non-hydrostatic (UA-ICON) model. Our results indicate that there are interhemispheric differences in the seasonal characteristics of the diurnal and semidiurnal tide. There also are some differences in the mean wind climatologies of the

10 models and the observations. Our results indicate that GAIA shows a reasonable agreement with the meteor radar observations during the winter season, whereas WACCM-X(SD) shows a better agreement with the radars for the hemispheric zonal summer wind reversal, which is more consistent with the meteor radar observations. The free running UA-ICON tends to show similar winds and tides compared to WACCM-X(SD).



# 1 Introduction

For space weather applications, there is a growing need for climatological boundary conditions of winds and temperature at the mesosphere/lower thermosphere (MLT) for climatological means as well as to assess the day-to-day variability due to atmospheric waves (Liu, 2016). In particular, the MLT as the transition region between the middle atmosphere and upper atmosphere is still not well understood, leaving some uncertainty in the forcing from below for the thermosphere and ionosphere. In the past decade, several general circulation models (GCMs) have been extended into the upper atmosphere such as GAIA (Jin et al., 2012) and WACCM-X(SD) (Liu et al., 2010a) to obtain an improved comprehensive understanding of the vertical coupling between the atmospheric layers. GAIA and WACCM-X have been cross-compared with other models (Pedatella et al., 2014b) or satellite observations (Pedatella et al., 2016; Liu et al., 2013). Recently, Borchert et al. (2019) completed a first 15 years long climatology run with UA-ICON with gravity wave parameterization. UA-ICON is the upper atmosphere extension of the non-hydrostatic ICON model of the German weather service and the Max Planck Institute for Meteorology (Zängl et al., 2015; Giorgetta et al., 2018).

GCMs can in general be run freely and develop their own meteorology, or in "nudged" mode by forcing their troposphere and or stratosphere to observed meteorology to constrain long-term climate simulations of the MLT to investigate potential long term changes e.g. due to the variable solar forcing within a solar cycle or other climate signals from below. In any case, an evaluation of the models with available information is required. In this study we present a climatological comparison of GAIA, WACCM-X(SD), and UA-ICON with ground-based meteor radar observations at mid- and polar latitudes on the northern and southern hemisphere. The study thus complements other studies investigating vertical coupling phenomena combining local ground-based observations with GCM data (Conte et al., 2017; Wu et al., 2019; Pancheva et al., 2020).

Long-term (over a solar cycle period) changes, or trends, observed in the thermosphere/ionosphere system are at least in part due to solar variability (e.g., Laštovička, 2017). The thermosphere trends could be in part attributed to the changes in lower atmosphere greenhouse gases (Emmert, 2015; Solomon et al., 2018). It is however under debate if the greenhouse effects are enough to explain the observed thermosphere/ionosphere trends in temperature and density, or direct thermodynamic effects from greenhouse gases also needs to be considered (Oliver et al., 2013; Zhang et al., 2016).

Ground-based observations provide also valuable observations to compare winds to GCM outputs. In the past Medium Frequency (MF) radars have been used to obtain MLT winds and to derive climatologies (Manson et al., 1989; Nakamura et al., 1993; Thorsen et al., 1997; Wilhelm et al., 2017). McCormack et al. (2017) validated the MLT winds of a meteorological analysis obtained with the Navy Global Environmental Model- High Altitude (NAVGEM-HA) and data from globally distributed meteor radars (MRs) for two winter seasons. They found a remarkable good agreement of the NAVGEM-HA winds and the MR observations even for time scales of days as well as for the tidal variability. Later, Stober et al. (2020a) extended the comparison to a full season using a smaller number of MRs on the northern hemisphere and included a cross-validation to a lidar temperature climatology. NAVGEM-HA assimilates also satellite observations up to the MLT (Eckermann et al., 2018)





and, thus, the agreement between the meteorological analysis and the MR winds provides confidence in both data sets. MR are widely used to observe MLT winds over meanwhile several decades making these instruments valuable assets to monitor climate variability and change in the MLT (Stober et al., 2014; Jacobi and Fytterer, 2012; Jacobi et al., 2015; Lilienthal and

Jacobi, 2015; Lukianova et al., 2018; Wilhelm et al., 2019).

In this study, we present a climatological comparison of MR winds and the corresponding GAIA, WACCM-X(SD) and UA-ICON fields for six meteor radars at conjugate mid- and polar latitudes to investigate interhemispheric differences and to evaluate how well the observations and the GCM data show similar dynamics. Therefore, we analyze the GAIA, WACCM-X(SD), and UA-ICON climate model runs and compile mean wind and tidal climatologies for the same periods as the available

MR measurements applying an Adaptive Spectral Filter (ASF). The ASF allows a harmonized and unified methodology to decomposing time series into mean winds and tidal information for both data sets (Baumgarten and Stober, 2019; Stober et al., 2020a).

The manuscript is structured as follows. Section 2 contains a description of the six meteor radar, as well as the GAIA, WACCM-X(SD) and UA-ICON data sets. In section 3 we present a brief description of the data analysis. The conjugate latitude compar-

ison for polar-latitudes is given in section 4 and for the mid-latitudes in section 5. Section 6 provides a comparison between high and mid-latitudes on the northern hemisphere. The results are discussed in section 7 and the conclusions are given in section 8.

## 2 Observations and models

### 2.1 Meteor radar observations

In this study we present long term observations of six globally distributed meteor radars located at Sodankylä (SOD) (67.9°N, 21.1°E), Kiruna (KIR) (67.4°N, 26.6 °E), Collm (COL) (51.3°N, 13.0°E), Tavistock (CMOR) (43.3°N, 80.8°W), Tierra del Fuego (TDF) (53.7°S, 67.7°W) and Davis (DAV) (68.6°S, 78.0°E). A detailed summary of each system can be found in Table 2.1. The systems are well-known and have proven to provide reliable and continuous measurements for cross-validation (McCormack et al., 2017; Stober et al., 2019) or long-term change studies (Iimura et al., 2011; Jacobi et al., 2015; Wilhelm

et al., 2019; Pancheva et al., 2020). In Figure 2.1, we present an overview of where each system is located. The meteor radar sites can be grouped in two conjugate geographic locations at mid- and polar latitudes. COL and TDF represent the mid-latitude conjugate observations and SOD and DAV for polar latitudes. KIR and CMOR are included as further validation setup to investigate the northern hemispheric latitudinal dependence in more detail.

Meteor winds are computed from the so called meteor position data (Hocking et al., 2001; Holdsworth et al., 2004). We

applied a harmonized data processing to generate homogeneous wind time series for all sites. The wind retrieval is described in more detail in (Stober et al., 2018) and was validated against meteorological analysis NAVGEM-HA (Navy Global Environment Model - High Altitude) (McCormack et al., 2015; Stober et al., 2019). NAVGEM-HA meteorological analysis utilizes a sophisticated 4DVAR data assimilation scheme, which assimilates observations including mesospheric data from MLS and SABER (Kuhl et al., 2013; McCormack et al., 2015; Eckermann et al., 2018). Given the remarkable agreement between

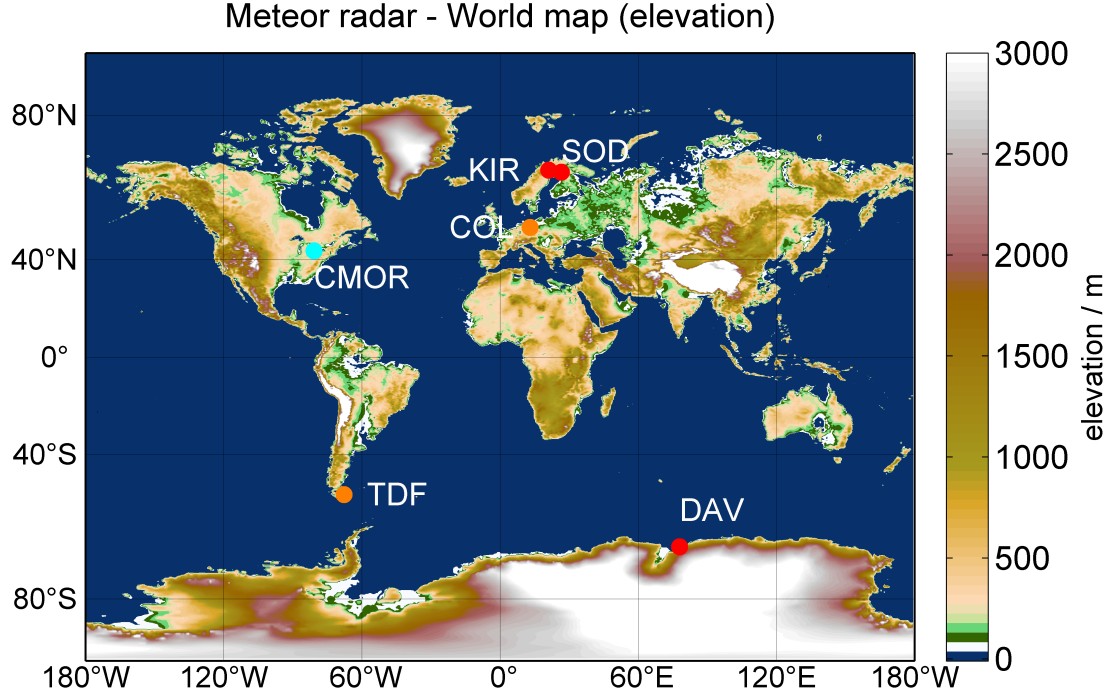

**Figure 1.** World map with meteor radar locations and color coded mean elevation. Conjugate latitude stations are indicated by the same colors. The plot was generated from etopo1 using the m_map package (Amante and Eakins, 2009).

NAVGEM-HA and the meteor radars winds for the general seasonal morphology as well as the short term variability, we consider the meteor radar winds as a proper validation reference for the WACCM-X, GAIA, and UA-ICON wind fields.

## 2.2  GAIA

GAIA is a 3-D self-consistent, fully-coupled whole atmosphere model of the Earth's troposphere, stratosphere, mesosphere,

thermosphere and ionosphere, covering the altitude range from the ground to ∼600 km for the neutrals and to 3000 km for the plasma (Jin et al., 2012). It has a horizontal resolution of 2.8° × 2.8° (latitude×longitude), and a vertical resolution of 0.2 scale heights. The model uses parameterizations to account for GWs, with formulations by McFarlane (1987a) for orographic GWs and by Lindzen (1981c) for non-orographic GWs. In the troposphere, stratosphere and mesosphere, a full radiation scheme developed by Nakajima et al. (2000) is used. The simulated atmosphere parameters (e.g. wind, temperature) are given



**Table 1.** Technical parameters of the meteor radars and experiment settings

|  | TDF | COL | SOD | KIR | DAV | CMO |
|---|---|---|---|---|---|---|
| Freq. (MHz) | 32.55 | 36.2 | 36.9 | 32.55 | 33.2 | 17.45, 29.85,38.15 |
| Power (kW) | 64 | 6/15 | 7.5/15 | 6 | 7 | 6/15/6 |
| PRF (Hz) | 625 | 2144/625 | 2144 | 2144 | 430 | 532 |
| coherent integration | 1 | 4/1 | 4 | 4 | 4 | 1 |
| pulse code | 7-bit Barker | mono/7-bit Barker | mono | mono | 4-bit complementary | mono |
| sampling (km) | 1.5 | 2/1.5 | 2 | 2 | 1.8 | 3 |
| location (lat,lon) | 53.7°S, 67.7°W | 51.3°N, 13.0°E | 67.4°N, 26.6 °E | 67.9°N, 21.1°E | 68.6°S, 78.0°E | 43.3°N, 80.8°W |
| observations | 02/2008-2020 | 08/2004-2020 | 12/2008-2020 | 12/1999-2020 | 02/2005-2020 | 01/2002-2020 |

in hourly values. GAIA has been demonstrated to be particularly good at capturing comprehensive coupling processes between the lower and upper atmosphere at different temporal and spatial scales, e.g, the wave-4 structure, the thermosphere cooling during stratosphere sudden warmings (SSWs) (Liu et al., 2009a; Liu et al., 2014).

This study uses the same 21-year long reanalysis data-driven simulation results as that used for ENSO study in Liu et al. (2017). Briefly, A nudging technique is used to constrain the model output (e.g. pressure, temperature, wind, etc.) below 30
km altitude to the reanalysis data JRA-25/55 by Japan Meteorological Agency with a $1.25° \times 1.25°$ spatial resolution and a 6-hour temporal resolution (Onogi and et al., 2007; Kobayashi et al., 2015). Due to the update of JRA-25 to JRA-55 in 2014, the simulation uses JRA-55 for 2014-2016 and JRA-25 before that. The F10.7 index as a proxy for the EUV input was set to observed values, while a fixed cross polar cap potential of 30 kV and a quiet particle precipitation condition were held throughout the simulation period to exclude any geomagnetic activity effect.

## 2.3  WACCM-X(SD)

Whole Atmosphere Community Climate Model Extension (WACCM-X) is one of the atmosphere configurations of the Community Earth System Model (CESM; Hurrell et al. (2013)). WACCM-X models the whole atmosphere from the lower boundary (representing ocean, land, or ice) to the upper boundary in thermosphere (500-700 km altitude depending on solar activity). Representation of the atmospheric physics in WACCM-X up to the lower thermosphere ($\sim$ 130 km altitude) is similar to that of
the conventional WACCM configuration (Marsh et al., 2013), while representation of the ionospheric electrodynamics is similar to the Thermosphere-Ionosphere Electrodynamics General Circulation Model (TIE-GCM; Richmond et al. (1992); Maute (2017)). Development and validation of the WACCM-X is described by Liu et al. (2018)[1].

---

[1]Details of the most recent release WACCM-X v2.1 can be found at https://www2.hao.ucar.edu/sites/default/files/users/whawkins/WxReleaseNotes2.1.pdf





The Specified Dynamics (SD/WACCM-X) simulation run deployed here (Gasperini et al., 2020) constrains tropospheric and stratospheric dynamics up to $\sim$ 50 km altitude using reanalysis based on the Modern-Era Retrospective Analysis for Research

and Applications (MERRA; Rienecker et al. (2011)). We refer to this run further on as WACCM-X(SD). The simulated atmospheric dynamics including zonal and meridional winds with 3 hour time resolution is given on the pressure levels with 1/4 scale height vertical resolution above the upper stratosphere, and uniform horizontal resolutions in latitude and longitude of 2.5° and 1.9°, respectively. The effects of non-orographic gravity waves (GWs) are parameterized using the source-orientated parameterization approach (Richter et al., 2010; Garcia et al., 2017). Orographic GWs are parameterized according to McFar-

lane (1987a). The external forcing due to varying geomagnetic activity is parameterized using the planetary Kp index with the high-latitude plasma convection specified according to Heelis et al. (1982).

## 2.4 UA-ICON

The Upper Atmosphere ICOsahedral Non-hydrostatic (UA-ICON; Borchert et al., 2019) atmospheric GCM covers the atmosphere from the surface to 150 km and is a vertical extension of the standard ICON configurations which have the lid usually at

about 80 km. ICON is available with different physics packages to be used for numerical weather prediction (Zängl et al., 2015) and climate studies (Giorgetta et al., 2018; Crueger et al., 2018). The upper atmosphere extension can be run with both physics packages. Here we make use of the latter. The upper atmosphere configuration extends the dynamical core from shallow- to deep-atmosphere dynamics and includes an upper atmosphere physics package with parameterizations for molecular diffusion, radiation in the Schumann-Runge bands and continuum, extreme UV, non-LTE effects and NO cooling, and chemical heating

(see Borchert et al. (2019) for details). The latter is necessary as UA-ICON does not calculate air chemistry interactively but uses prescribed climatologies of radiatively active species. Furthermore, UA-ICON uses all physics parameterizations from the standard configuration as described by Giorgetta et al. (2018), including in particular parameterizations for effects from non-orographic gravity waves (Hines, 1997) and sub-grid scale orography (Lott, 1999). ICON uses a triangular horizontal grid derived from a spherical icosahedron by subdivisions of triangular cells. Here we are making use of a so-called R2B4 grid

with a horizontal resolution of about 160 km. In the vertical, ICON uses a terrain following hybrid sigma height grid with 120 layers in this case. Rayleigh damping of vertical winds is applied above 120 km. The simulation presented here is using the same model configuration as the climatological test case described by Borchert et al. (2019). It has been run with climatological present-day like boundary conditions for 21 years with the first year discarded as spin-up. In contrast to the GAIA and WACCM-X(SD) experiments, the meteorology in the UA-ICON simulations has developed freely as no nudging technique has

been applied.

## 3  Data analysis of mean winds and atmospheric tides

Comparing model fields and observations is very often not as straightforward as expected as the model data has usually a different spatial and temporal resolution as the observations. Thus, in a first step we performed a data reduction to the WACCM-X(SD), GAIA, and UA-ICON global data sets by cutting out all grid points in the vicinity of the meteor radars



using a 300 km radius, which is a bit more than the actual beam width used in the wind retrieval of about 220 km radius, but
       ensures that at least 5 grid points are available from each model and for each site. We extracted for each meteor radar location
       the geopotential height, the zonal and meridional wind, temperature as well as pressure for all grid points that fall within the
       above mentioned area around the meteor radars. These reduced data sets are now further analyzed to simulate meteor radar
       observation. Therefore, we converted the geopotential heights ($\Phi$) into geometric altitudes ($h$) for each extracted profile using
the expression by taking into consideration of variable gravity;

$$h(\text{lat},\text{lon}) = \Phi(\text{lat},\text{lon})/\left(1 - \frac{\Phi(\text{lat},\text{lon})}{R_{\text{Earth}}(\text{lat},\text{lon})}\right). \tag{1}$$

Here $R_{Earth}(lat,long)$ corresponds to the Earth radius at a given latitude and longitude. In a next step, the converted height
vectors for each profile are interpolated to a reference altitude vector, which has a vertical resolution of 2 km between 16-150
km, 5 km vertical resolution at altitudes from 155-200 km and 10 km vertical resolution between 210-300 km and a 20 km
vertical resolution at altitudes above 320 km to account for the decreased model resolutions due to the pressure level spacing
       in the models. Finally, we compute the median and variance for all profiles and obtain a mean zonal and meridional wind
       and temperature corresponding to the observation volume of each meteor radar. Furthermore, we derive the variance of these
       parameters, which provides a proxy for the statistical uncertainties similar to the meteor radar observations. The final result of
       our data reduction are time series of zonal and meridional winds with a temporal resolution of 1 hour for GAIA and UA-ICON
and 3 hours for WACCM-X(SD) respectively, and a 2 km vertical resolution at the mesosphere/lower thermosphere, which is
       identical to the meteor radar observations.

MR winds are obtained using the retrieval algorithm described in Stober et al. (2018), which is basically a further evolution
of Hocking et al. (2001) and Holdsworth et al. (2004). The retrieval includes a full Earth geometry based on the WGS84
reference ellipsoid, full non-linear error propagation and a spatio-temporal Laplace filter as Tikhonov regularization constraint.
Furthermore, the wind retrieval does not require $w = 0$ and explicitly fits for the vertical component, which are considered as
       remaining wind bias due to the lack of independent validation sources. However, these vertical winds have proven to provide
       a good quality control and show a Gaussian distribution with a width of $w \pm 0.25$ to $\pm 0.35$ m/s around the zero wind line.
       The benefit of this retrieval is that we obtain for all systems a harmonized wind time series based on the same quality control
       criteria.

Atmospheric mean winds and tides are analyzed using the ASF, which is described in more detail in Baumgarten and Stober
       (2019) and Stober et al. (2020a) and was already applied in several studies (Stober et al., 2017; Pokhotelov et al., 2018; Wilhelm
       et al., 2019; Stober et al., 2020b) to decompose MR winds in daily mean winds, diurnal and semidiurnal tides for the zonal
       and meridional components, respectively. The technique is implemented based on least square fits with full error propagation,
       which permits to apply the algorithm to unevenly sampled data with data gaps. Similar to wavelets the window length is
adapted for each of the fitted wave periods (Torrence and Compo, 1998). Furthermore, we minimize the impact of inertia-scale
       gravity waves on the tidal analysis by applying a vertical regularization to the tidal phases. In Stober et al. (2020b) shows
       an example comparing the ASF2D with classical harmonic analysis for different window lengths. Due to the intermittent and
       non-stationary wave field generated by gravity waves and tides, long window lengths tend to produce artefacts and leak energy





between the different wave scales. Furthermore, the meteor radar sampling is irregular in time, which additionally introduces
spectral leakages that are significantly reduced by the ASF2D technqiue.

The GAIA, WACCM-X(SD), UA-ICON and MR time series are analyzed with the same ASF2D algorithm to ensure the best
possible comparability and to minimize differences, which might be introduced when different analysis procedures are applied.
Thus, we obtain harmonized time series for daily mean winds, diurnal and semidiurnal tidal amplitudes and phases as well as
a gravity wave spectral residuum for each data set. The model data is available with different temporal resolutions, which
refers to the cadence of the data output of the meteorological fields rather than the actually numerical temporal step size (e.g.
WACCM-X(SD) is solved with 5 minutes resolution). Hence, the model data for each temporal step represents the numerical
solution for this output period and not a temporal average as in the observation. Furthermore, we performed an additional test
to ensure that the coarser temporal resolution of WACCM-X(SD) of 3 hours has no impact on the harmonized time series.
Thus, we used an earlier version WACCM-X (v.1.9) run with hourly data output for cross-validation and found no resolution
dependent issues.

Vertical wavelengths of the diurnal and semidiurnal tides are also estimated from the vertical phase profiles. Therefore, we
estimated the instantaneous vertical linear slope of the unwrapped phases in the altitude range between 80-100 km. The vertical
wavelength is then estimated from the altitude difference between the $-\pi$ and $\pi$ phase transitions. This method allows to derive
vertical wavelengths that are much longer than the actual width of the meteor layer. Such long vertical wavelengths correspond
to evanescent tidal modes. However, we did not define a certain threshold, but consider vertical wavelengths that are much
longer than 300 km as evanescent or as not vertically propagating. Vertical wavelengths of the diurnal tide were truncated
at 1000 km for plotting reasons. Semidiurnal tides only occasionally showed such long wavelengths and, thus, are presented
without truncation.

## 4 Conjugate comparison at polar latitudes

### 4.1 Mean winds


SOD and DAV are located at conjugate latitudes in Arctic and Antarctic sectors, respectively. Figure 2 shows a comparison of
the meteor radar observations of zonal and meridional daily mean winds. Panel a) presents the zonal component for SOD (left
column) and DAV (right column). The upper row visualizes the meteor radar measurements, the 2nd row shows GAIA, the 3rd
row shows WACCM-X(SD), while the lower row shows UA-ICON.
Zonal winds exhibit a characteristic seasonal pattern with weak eastward winds during the hemispheric winter, a wind reversal
from westward to eastward winds with a gradual change of the reversal altitude over the hemispheric summer months. Fur-
thermore, there is a characteristic asymmetry of the spring transition compared to the fall. The SOD MR observes westwards
winds mid-March to May at altitudes up to 100 km, whereas the corresponding structure at DAV shows near zero wind at 100
km in October-November and a much stronger westward wind enhancement at approximately 90 km altitude.
GAIA reproduces some features of the seasonal morphology of the zonal wind. Mainly, the model shows eastward winds dur-
ing the hemispheric winter season with a similar magnitude as the MR measurements at SOD and DAV. However, the summer





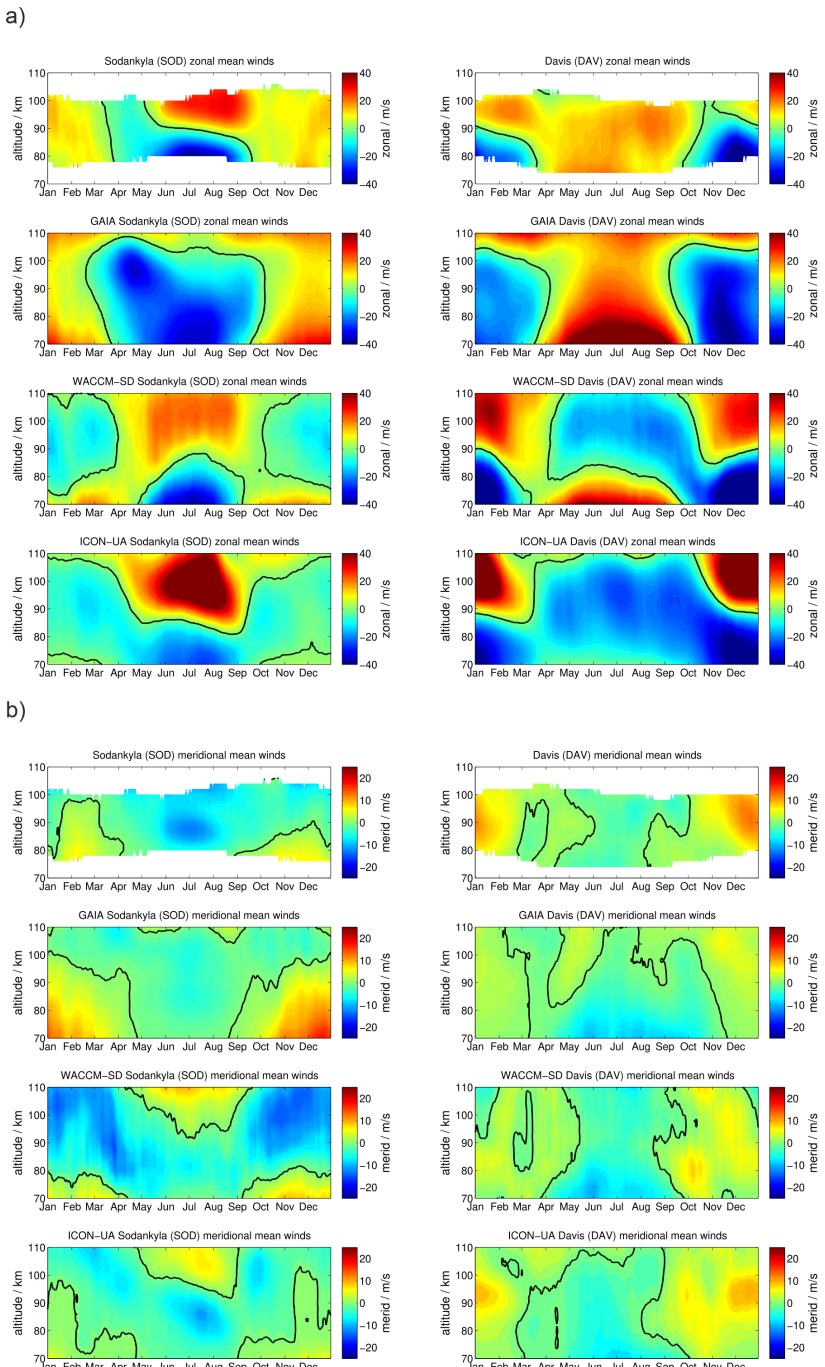

**Figure 2.** Comparison of zonal (panel a) and meridional (panel b) daily mean winds for the MRs at SOD (left column) and DAV (right column).





wind reversal occurs at altitudes 10-15 km higher compared to the observations. Furthermore, GAIA does not indicate an eastward wind enhancement in the shown altitude range. Another interesting feature of GAIA is that the seasonal asymmetry of the spring and fall transition is visible in the model data. Interhemispheric differences are also shown by the model. The winter
eastward wind magnitudes are about 5-8 m/s increased at southern polar latitudes compared to the conjugate location in the northern hemisphere.

The seasonal morphology in WACCM-X(SD) shows substantial differences from the MR measurements during the hemispheric winter months: the zonal wind changes from eastward to westward between 70-80 km and remains westward at most MLT heights, whereas observational data shows no reversal and are eastward in this region. The summer wind reversal from
westward to eastward winds can be found in the model as well as in the observations. As such, the general seasonal morphology tends to be more symmetric in WACCM-X(SD). The spring and fall transitions look very similar. The summer mesospheric wind reversal does not exhibit the gradual descent of the reversal altitude. Southern hemispheric winds at DAV show increased magnitudes compared to the observations. The zonal wind pattern is thus not well-represented during winter months in both hemispheres for SOD and DAV in WACCM-X(SD). Similarly to WACCM-X(SD), UA-ICON does not reproduce westward
winds during the winter months. It also indicates even stronger eastward winds on both hemisphere during the summer months. Furthermore, it is noticeable that UA-ICON tends to capture the seasonal asymmetry in the zonal winds and shows a gradual decrease of the summer wind reversal altitudes as it is seen in the observations.

Meridional winds are compared in Figure 2 panel b). MRs at conjugate latitudes are supposed to see almost the same qualitative seasonal pattern without any phase shift. SOD and DAV show northward winds during northern hemispheric winter and
southward winds during the summer reflecting the hemispheric upwelling above the summer pole and the downwelling during the winter months.

GAIA exhibits a very similar seasonal characteristic for both stations. However, the northward winds during the northern hemispheric winter have an increased magnitude compared to the observations at SOD and are less strong above DAV. Nevertheless, GAIA is capable of capturing the main seasonal features in the meridional component for both locations.
Comparing meridional winds in WACCM-X(SD) with the observations reveals distinct differences between the conjugate latitudes. At the southern hemisphere above DAV the seasonal morphology is well-reproduced in WACCM-X(SD) and shows the northward winds during the northern hemispheric winter and southward winds during May to August. This is not the case for SOD, where WACCM-X(SD) shows an entirely different seasonal meridional wind throughout the year, which is most of the time southward at the altitude range between 80-100 km. Furthermore, the model exhibits a wind reversal from southward to
northward during the summer months May to September above 100 km, which is not indicated in the observations. Meridional winds in UA-ICON show again a seasonal characteristic similar to WACCM-X(SD). However, the magnitude of the meridional winds appears to be in better agreement with the observations. In particular, the meridional winds at DAV look similar compared to the observations.





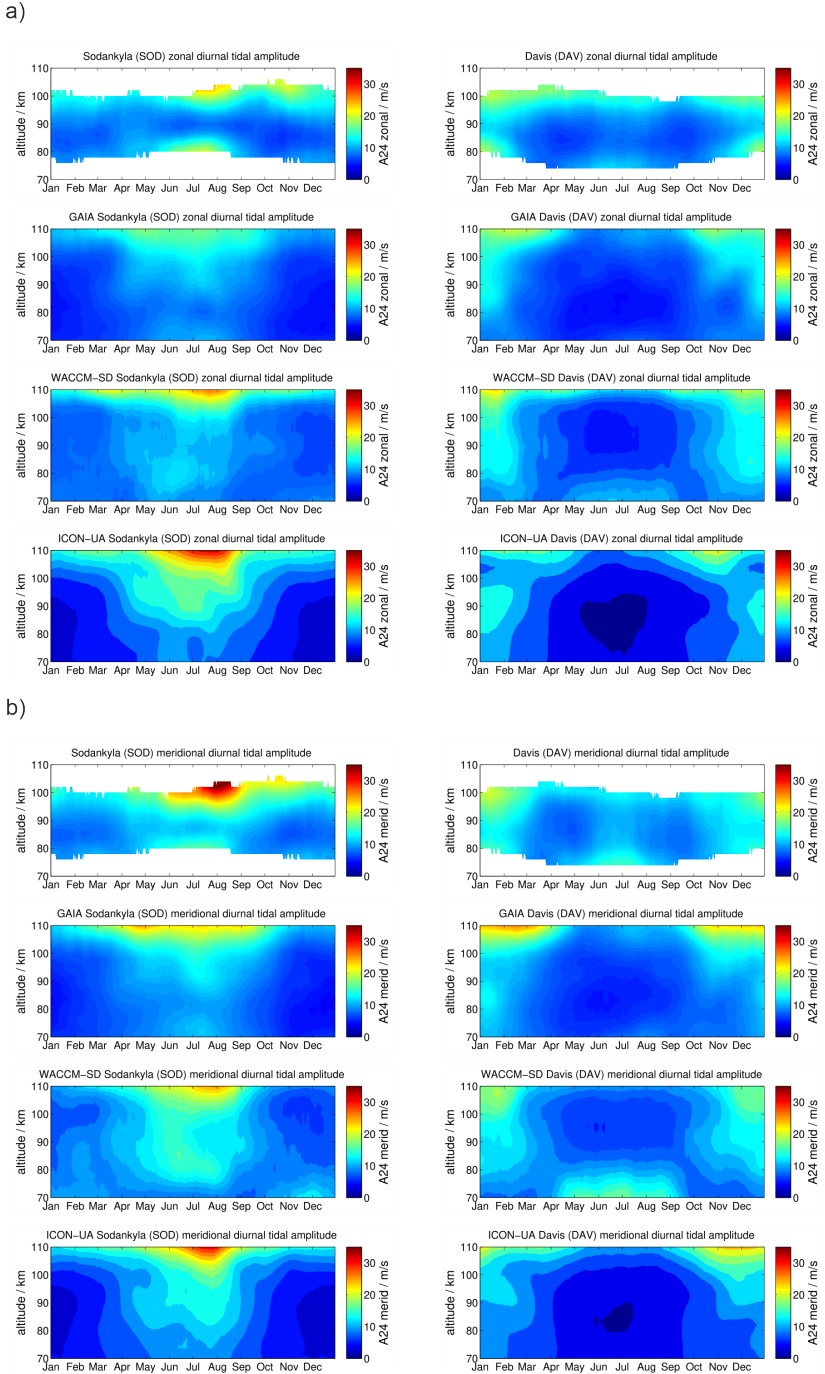

**Figure 3.** The same as Figure 2, but for the diurnal tidal amplitudes.

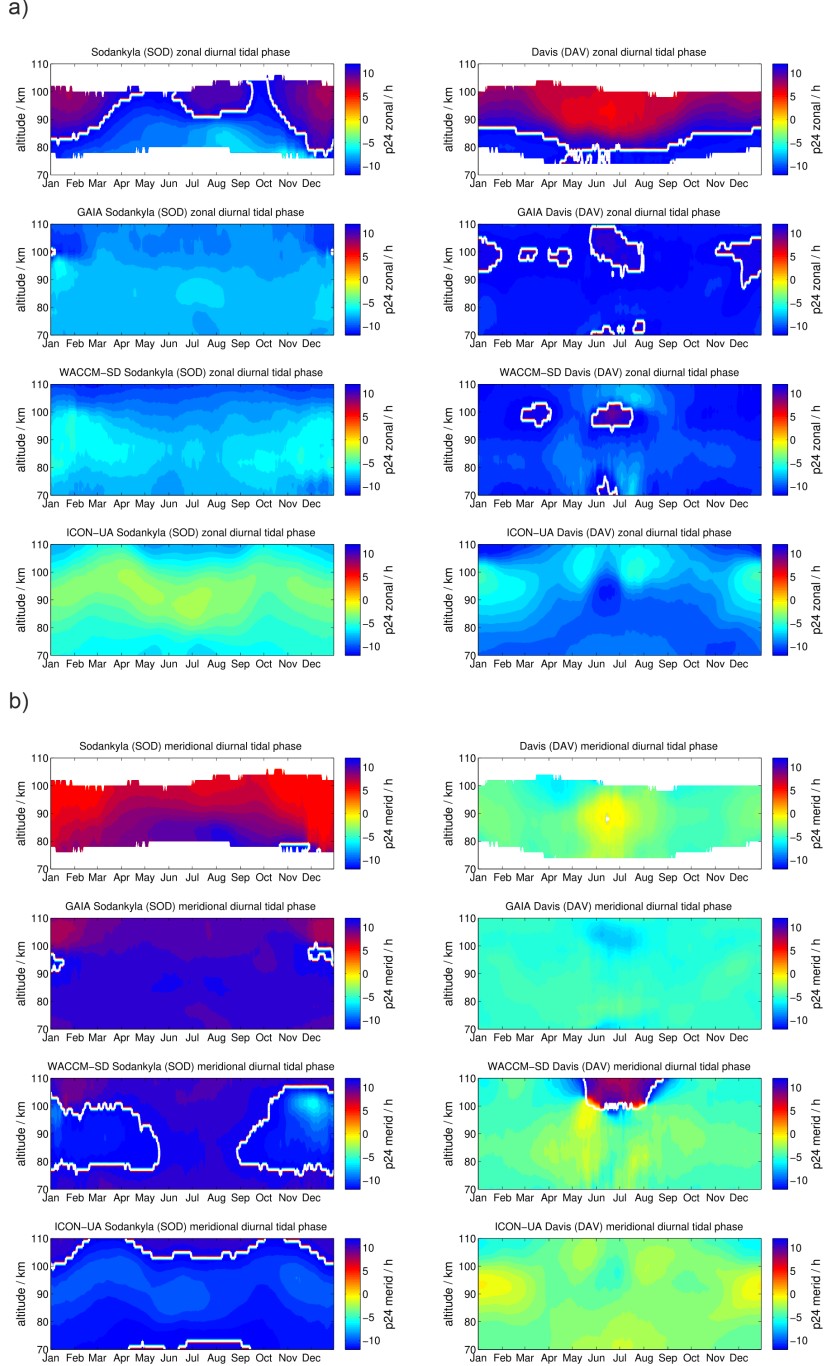

**Figure 4.** The same as Figure 3, but for the diurnal tidal phases. The white line indicates a phase jump or the zero phase transition.





## 4.2 Diurnal tides

At polar latitudes diurnal tides gain only moderate amplitudes, although being still visible throughout the year, which is also predicted from the Laplace tidal equation and the corresponding Hough modes (Andrews et al., 1987; Wang et al., 2016). The amplitudes reach their largest values during the hemispheric summer months. Furthermore, the zonal and meridional amplitudes show consistent seasonal patterns. Typically, at middle and high latitudes the meridional amplitudes exceed the zonal amplitudes during the summer months as documented before (Portnyagin et al., 2004; Jacobi, 2012; She et al., 2016; Wilhelm

et al., 2017; Baumgarten and Stober, 2019; Pancheva et al., 2020). Figure 3 presents the comparison between SOD and DAV with a similar arrangement of the panels as for the mean winds. The MR observations reveal a characteristic vertical structure of the diurnal tidal amplitude for the hemispheric summer months, which shows a first enhancement at altitudes below 80 km and a second maximum above 95-100 km. Furthermore, there is a second hemispheric winter maximum apparent at DAV during June and July at altitudes below 80 km, which is less obvious at SOD.

GAIA and UA-ICON capture most of the seasonal characteristic of the diurnal tidal amplitudes above 90 km, but shows no tidal enhancements below 80 km. The meridional amplitudes are also more amplified compared to the zonal component at both hemispheres, which is consistent to the observations. However, the vertical structure of diurnal amplitudes is less visible relative to the observations. This is also the case for WACCM-X(SD). WACCM-X(SD) also shows the diurnal tidal enhancement above DAV during the hemispheric winter below 80 km, which is not found in GAIA and UA-ICON. However, in WACCM-

X(SD) this lower amplitude maximum is stronger than amplitudes above 100 km during the same period, which appears to be reversed compared to the observations. In general the amplitudes of the diurnal tide appear to be almost equal in the zonal and meridional component for DAV, which is also reflected by all three models. Only at SOD the observations indicate enhanced amplitudes in the meridional component, which seems to be less pronounced in all three model outputs. One noticeable aspect of UA-ICON is the diurnal tidal seasonal climatology, which indicates much lower amplitudes during hemispheric winter

compared to the observations and the other two models.

Comparing the diurnal tidal phase between SOD and DAV MR observations we found remarkable differences in the seasonal pattern between both hemispheres (see Fig. 4). The zonal and meridional diurnal phase at DAV remains more or less stable throughout the year indicating only little annual variation with longer vertical wavelengths in the meridional component, whereas at SOD a pronounced semiannual structure is observed with distinguished phase changes in April/May and October.


WACCM-X(SD) and GAIA present a very similar seasonal diurnal tidal phase characteristic for both components, which deviates by several hours relative to the MR measurements at both locations. In this respect, there is a less good agreement of the vertical diurnal tidal phase structure for both models in comparison to the observations. For the DAV location WACCM-X(SD) meridional phase even shows a jump above 100 km, which is not seen in GAIA. UA-ICON diurnal tidal phases exhibit

an offset compared to the observations, but the vertical structure appears to be similar to GAIA and WACCM-X(SD).

Vertical wavelengths for the diurnal tide are presented in Figure 5. All three models tend to overestimate the vertical wavelength compared to the MR observations, which could be already seen in the phase plots. These very long vertical wavelengths





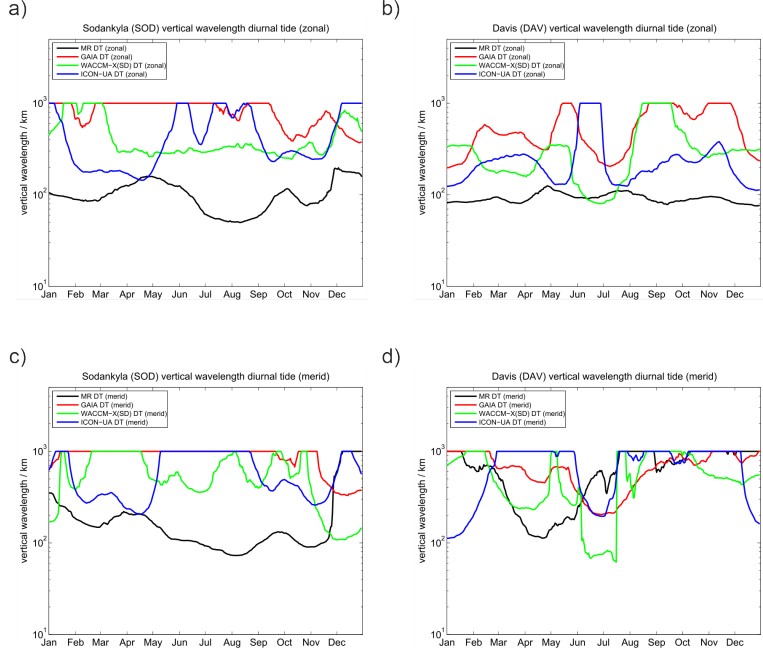

**Figure 5.** Comparison of vertical wavelengths of the diurnal tide. The left column shows in panel a) the zonal and panel c) meridional vertical wavelength for SOD. Panel b) and d) present the same but for DAV.

reaching more than 1000 km n all models during certain periods practically indicate that the tidal modes are evanescent and not vertically propagating.

### 4.3 Semidiurnal tides

Semidiurnal tides are the dominating tides at mid- and high-latitudes at the MLT and reveal a characteristic seasonal structure (Portnyagin et al., 2004; Wilhelm et al., 2019; Pancheva et al., 2020; van Caspel et al., 2020), but also show significant inter-hemispheric differences. Figure 6 compares the semidiurnal amplitudes. The panels are arranged as for the diurnal tides. At the location of SOD in the northern hemisphere the semidiurnal tidal amplitudes are largest during the winter and autumnal equinox in September. The tides gain already values of more than 20 m/s at 90 km altitude and grow further with increasing altitude up to 35 m/s. During spring and summer months the semidiurnal amplitudes are smallest at SOD. In the southern hemisphere the seasonal characteristic is apparently different. DAV also shows a maximum of the semidiurnal tide at hemispheric autumnal equinox (April-May), but they remain rather weak during the hemispheric winter from June to August. In contrast to the northern hemisphere the semidiurnal tide enhances again during the hemispheric spring transition from September to December at altitudes above 90 km.

This much more complicated seasonal behaviour is only partly reproduced in GAIA, WACCM-X(SD), and UA-ICON. The models show increased semidiurnal tidal amplitudes during the winter months at SOD and DAV with the largest amplitudes





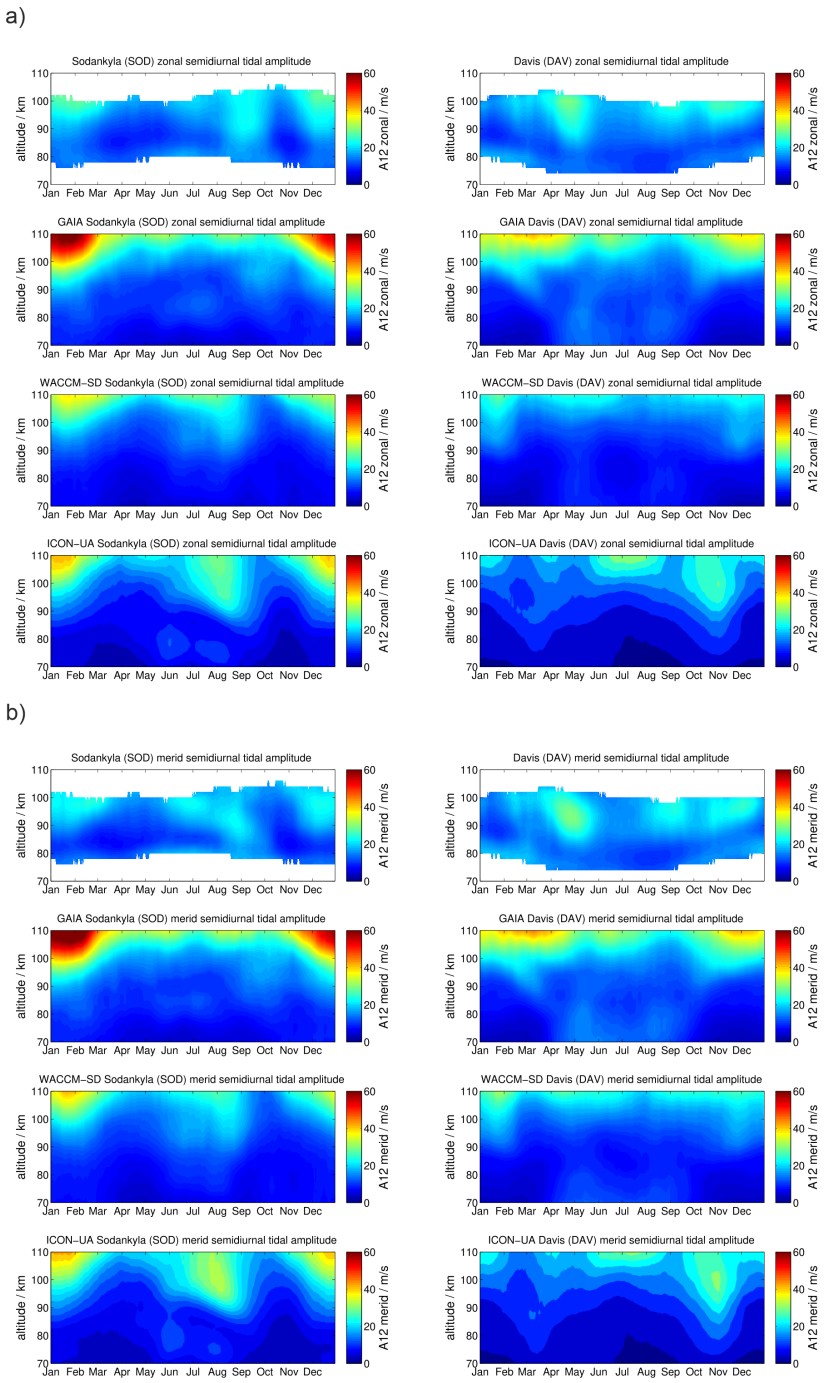

**Figure 6.** The same as Figure 2, but for the semidiurnal tidal amplitudes.





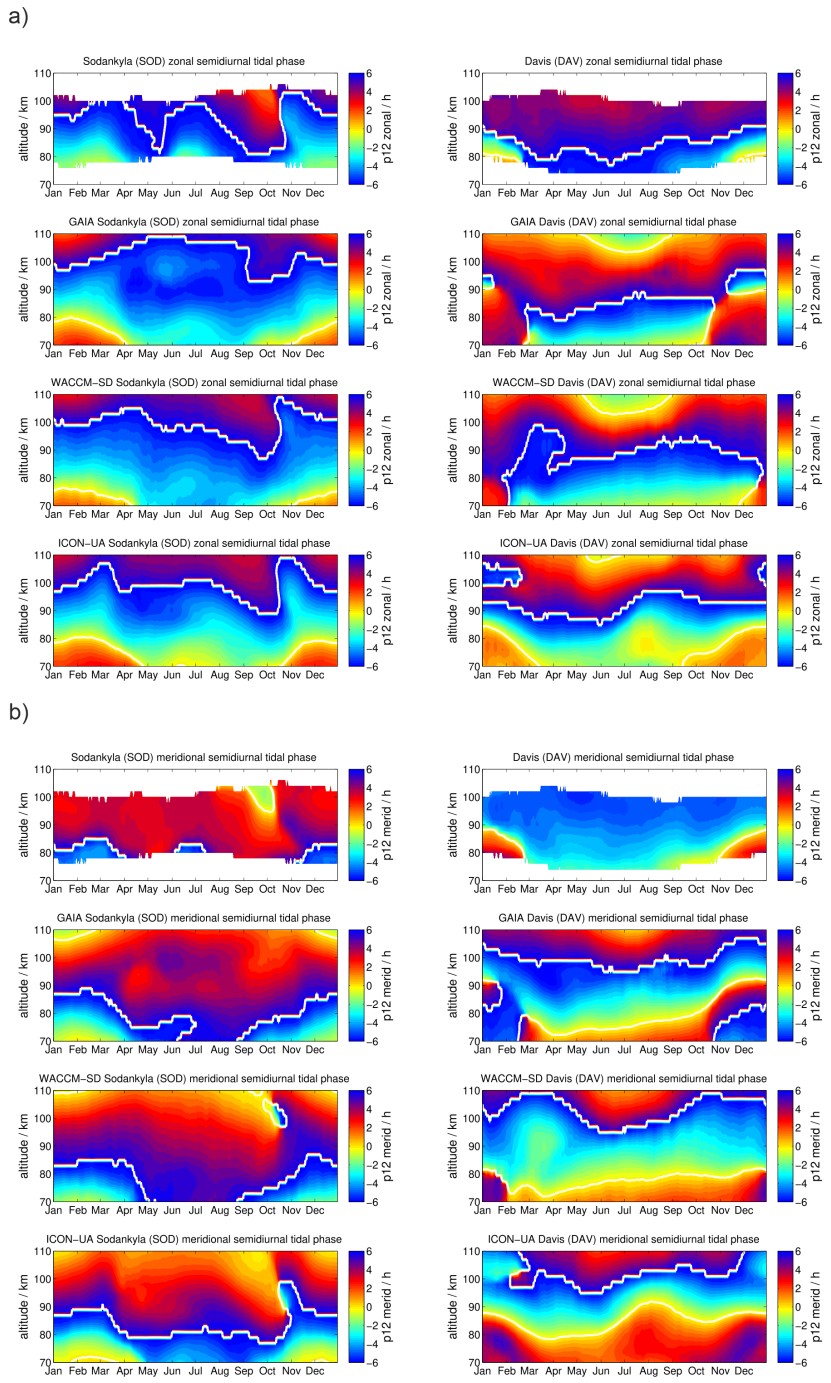

**Figure 7.** The same as Figure 2, but for the semidiurnal tidal phases.





reached in GAIA above 100 km altitude. The semidiurnal enhancement during the autumnal equinox is basically not present in GAIA in either hemisphere. WACCM-X(SD) indicates an enhancement of the tide for the northern hemisphere in both com-

ponents during August, which is about one month too early compared to the MR observations at SOD. UA-ICON indicates a similar seasonal characteristic as WACCM-X(SD) and the observations. The general seasonality of the semidiurnal tidal amplitudes is fairly consistent between the models, but exhibits deviations from the observations, which are more significant in the southern hemisphere.

In Fig. 6 we present the semidiurnal tidal phases for SOD and DAV. As already shown for the amplitudes, there are significant

interhemispheric differences at polar latitudes. The SOD MR observes a quasi-biannual tidal phase characteristic, whereas at DAV station the annual pattern is more evident. In the northern hemisphere the semidiurnal tidal phase is basically drifting with season and shows phase variations of several hours through the course of the year. Southern polar latitudes exhibit a much smaller phase variability and more constant seasonal variation of the phase, which is still significant and exceeds several hours, but the changes are not as rapid as in the northern hemisphere.

As expected from the diurnal tidal phase analysis, the models have difficulties to match the phase variability and seasonal characteristics especially in the southern hemisphere. GAIA appears to have a slightly better agreement with the observations above DAV at altitudes between 80-100 km in both components. WACCM-X(SD) indicates a better performance for the northern hemisphere and, in particular, captures the phase variability during the fall transition in both components, which is only weakly present in the GAIA data. Surprisingly (taking into account that the model is free-running), UA-ICON semidiurnal

tidal phases are reasonable well-reproducing the observations for the northern hemisphere and partly also in the southern hemisphere. Similar to WACCM-X(SD), UA-ICON exhibits the rapid phase changes during the northern hemispheric fall transition and shows a bit a more pronounced phase variability during the spring transition.

Finally, a comparison of vertical wavelengths of the semidiurnal tide at SOD and DAV is shown in Figure 8. The first noticeable aspect is the interhemispheric difference of the vertical wavelength between SOD and DAV MR observations. The

vertical wavelength appears to be much more variable in the course of the year in the northern hemisphere and takes values of about 50-60 km for the winter and prominent enhancements during April-June and October-November with values beyond 1000 km. Such large vertical wavelengths point to actually evanescent tidal modes, which essentially are not vertically propagating. From July to September vertical wavelengths of about 100-120 km are observed with the SOD MR. The behaviour at the southern hemisphere is rather different. During hemispheric summer, short vertical wavelengths of about 25 km are seen

above DAV station, which increase to about 100-120 km during March/April and gradually decrease again afterwards. The very long vertical wavelengths that are found above SOD are missing in the southern hemisphere. Similar to the semidiurnal tidal phase, GAIA shows a better agreement with the observations at the southern hemisphere during the hemispheric winter season from March to October. However, GAIA exhibits two distinct peaks in the vertical wavelength taking values beyond 1000 km in February and October/November, which are not found in the MR observations at DAV. At the northern hemisphere

the vertical wavelength seasonal climatology of GAIA is reasonably well-reproduced and the main features such as the triple peak structure are visible, but the model underestimates the vertical wavelengths. WACCM-X(SD) and UA-ICON show some interhemispheric differences and capture remarkably well the fall transition at the northern hemisphere, but underestimate the





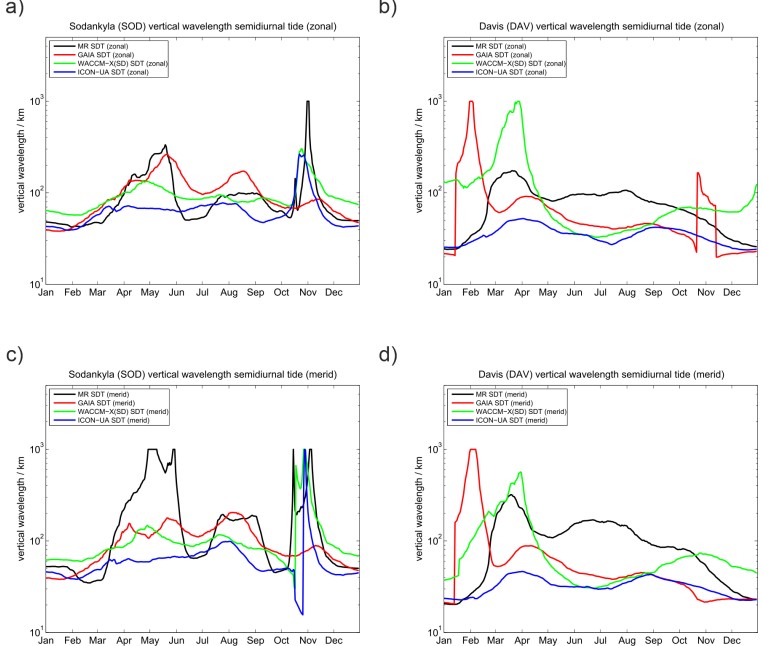

**Figure 8.** The same as Figure 5, but for the semidiurnal tidal vertical wavelength.

vertical wavelengths during April/May and July/August. Vertical wavelengths at the DAV station are also indicating some systematic deviations from the MR measurements. UA-ICON apparently underestimates the vertical wavelengths compared to the

observations and other models during March/April in both hemispheres, but indicates otherwise a remarkable similarity to the observations and WACCM-X(SD).

## 5   Conjugate comparison at mid-latitudes

### 5.1   Mean winds

Now we are comparing the mid-latitude stations COL and TDF. Although both stations are at almost conjugate latitudes,

they are located at very different geographic regions, which have to be considered. The COL MR station is in the center of Europe and far away from significant orographic mountain wave sources. The Alps are more than 600 km to the southwest and the Scandinavian mountains are almost 1000 km to the northwest. The TDF MR is much closer to the Andes mountains in Patagonia, which is one of the most relevant mountain wave hot spots on Earth and, thus, the observations are directly affected by the mountain wave source and a potential secondary wave forcing up to the mesosphere (Becker and Vadas, 2018; Vadas

and Becker, 2018).

Figure 9 presents the zonal and meridional daily mean wind climatologies for COL and TDF. As already shown in previous studies (Pokhotelov et al., 2018; Wilhelm et al., 2019), the main difference to the polar latitudes is found in a lower altitude of





a)

b)

**Figure 9.** The same as Figure 2, but for the mid-latitudes at COL and TDF.



the summer zonal wind reversal in both hemispheres and a corresponding altitude shift of the meridional southward/northward winds. However, there are some distinct interhemispheric differences between both locations and wind components. During the

spring and fall transitions zonal winds above 90 km show characteristic differences. TDF observations exhibit eastward winds throughout the year and indicate a weakening of them during the spring and fall period, whereas at COL from March/April until May and in October there is a wind reversal to weak westward winds. Meridional winds have a very similar seasonal characteristic due to the conjugate location of both stations. The meridional wind is always stronger above the stations in the summer hemisphere. Furthermore, the vertical structures are different. The COL MR shows a remarkable vertical structure of

meridional southward wind magnitude for the summer months from June to August, which exhibits the highest magnitudes at altitudes corresponding to westward zonal wind. In the southern hemisphere, we observe a smoother vertical structure of the meridional wind, which even reverses to northward above 84 km in July and August.

As already found for the polar latitudes, the zonal wind climatologies are only to a certain extent reproduced in the models. GAIA shows a better agreement during the winter months, whereas WACCM-X(SD) and UA-ICON exhibit a better agreement

of the summer zonal wind reversal compared to the MR measurements. All three models tend to overestimate the strength of zonal summer wind systems. The westward winds are too strong and extend to too high altitudes in GAIA. WACCM-X(SD) and UA-ICON indicate strong zonal wind reversals at the altitude where these are found in the observations, but the westward and eastward winds tend to be overestimated. Furthermore, UA-ICON indicates the seasonal asymmetry in the zonal wind, which is a less visible in WACCM-X(SD). GAIA meridional winds and the observations are in better agreement in the

southern hemisphere compared to the northern mid-latitudes concerning their strength and seasonal characteristic. Only the meridional wind reversal in July/August at TDF is not found in the GAIA data. At the northern mid-latitudes GAIA shows a more pronounced seasonal structure and underestimates the strength of the meridional southward wind during the summer months, which is likely associated with the summer zonal wind reversal that is not visible in GAIA. In winter, WACCM-X(SD) and UA-ICON exhibit more remarkable differences relative to GAIA and the observations. In the northern hemisphere the

meridional wind is southward at altitudes between 90 and 100 km throughout the year. Both models apparently only reproduce the seasonality, that is found in the observations, below 80 km altitude. In the southern hemisphere WACCM-X(SD) and UA-ICON show a better agreement during the hemispheric summer months (Dec-Feb), but tend to overestimate the wind reversal from southward to northward as well as the seasonal duration compared to the TDF observations.

## 5.2 Diurnal tides

At mid-latitudes diurnal tidal amplitudes have a similar seasonal characteristic as at the polar latitudes. During the hemispheric summer season the amplitudes are largest and remain rather small throughout the other months. Figure 10 shows a comparison of the diurnal tidal amplitudes between COL and TDF. Diurnal tidal amplitudes exhibit only small interhemispheric differences for the mid-latitudes. The highest amplitudes are observed above 100 km and reach values of 15-20 m/s for the zonal component and about 25 m/s for the meridional component. There is a good agreement between the observations in both hemispheres and

GAIA, WACCM-X(SD) and UA-ICON. All three models reproduce the seasonality of the amplitudes and partly of their vertical structure.





a)

b)

**Figure 10.** The same as Figure 2, but for the diurnal tidal amplitudes above COL and TDF.





a)

b)

**Figure 11.** The same as Figure 2, but for the diurnal tidal phases above COL and TDF.



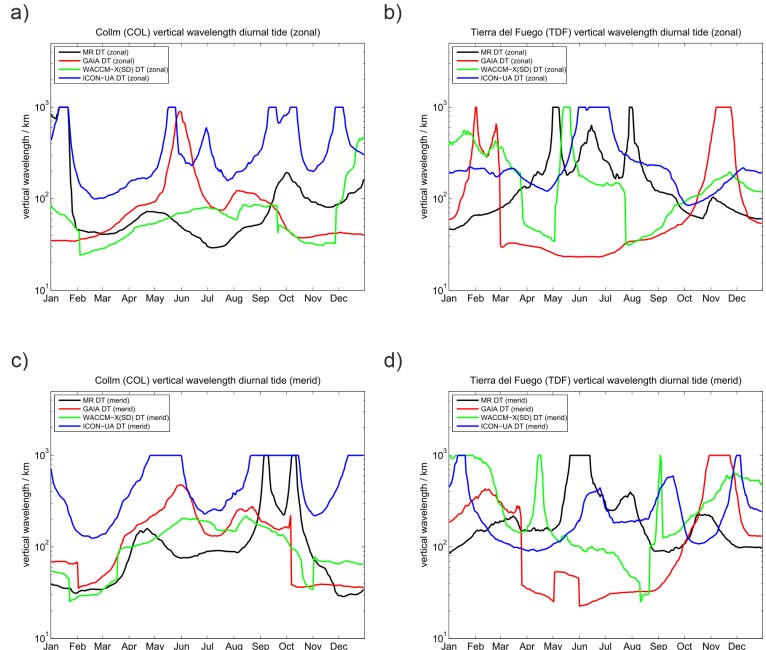

**Figure 12.** The same as Figure 5, but for COL and TDF.

Diurnal tidal phases are presented in Figure 11. Some of the features found at the polar latitudes are again visible at the mid-latitudes. The seasonal characteristic of the diurnal tidal phases indicates some interhemispheric differences, which were already indicated for SOD and DAV. In the northern hemisphere the tidal phase shows a more pronounced biannual mode
at altitudes above 84 km, whereas in the southern hemisphere an annual structure is more evident. All three models show dissimilarities in the seasonal vertical phase characteristic. Above 95 km GAIA and WACCM-X(SD) systematically deviates from the observations. UA-ICON exhibits an offset in the diurnal tidal phases compared to the observations as well as GAIA and WACCM-X(SD). In general, below 95 km, there is a much better agreement between the observations and models.

Figure 12 shows the vertical wavelength for the diurnal tide at the mid-latitude locations above COL and TDF. All three
models and the observations look dissimilar. Interestingly, also an inter-model comparison does not indicate a substantial agreement for the vertical wavelengths between the different models. However, contrary to the polar latitudes UA-ICON now tends to overestimate the vertical wavelength a bit more compared to GAIA and WACCM-X(SD), which was opposite at polar latitudes.

### 5.3 Semidiurnal tides

At mid-latitudes the semidiurnal tide is the dominating atmospheric wave and gains amplitudes of about 50 m/s on an average and occasionally up to 70 m/s above 90 km altitude. The seasonal characteristic is presented in Figure 13 and exhibits an inherent interhemispheric difference. The seasonal amplitude behaviour at COL looks rather similar to that at SOD, but reflects





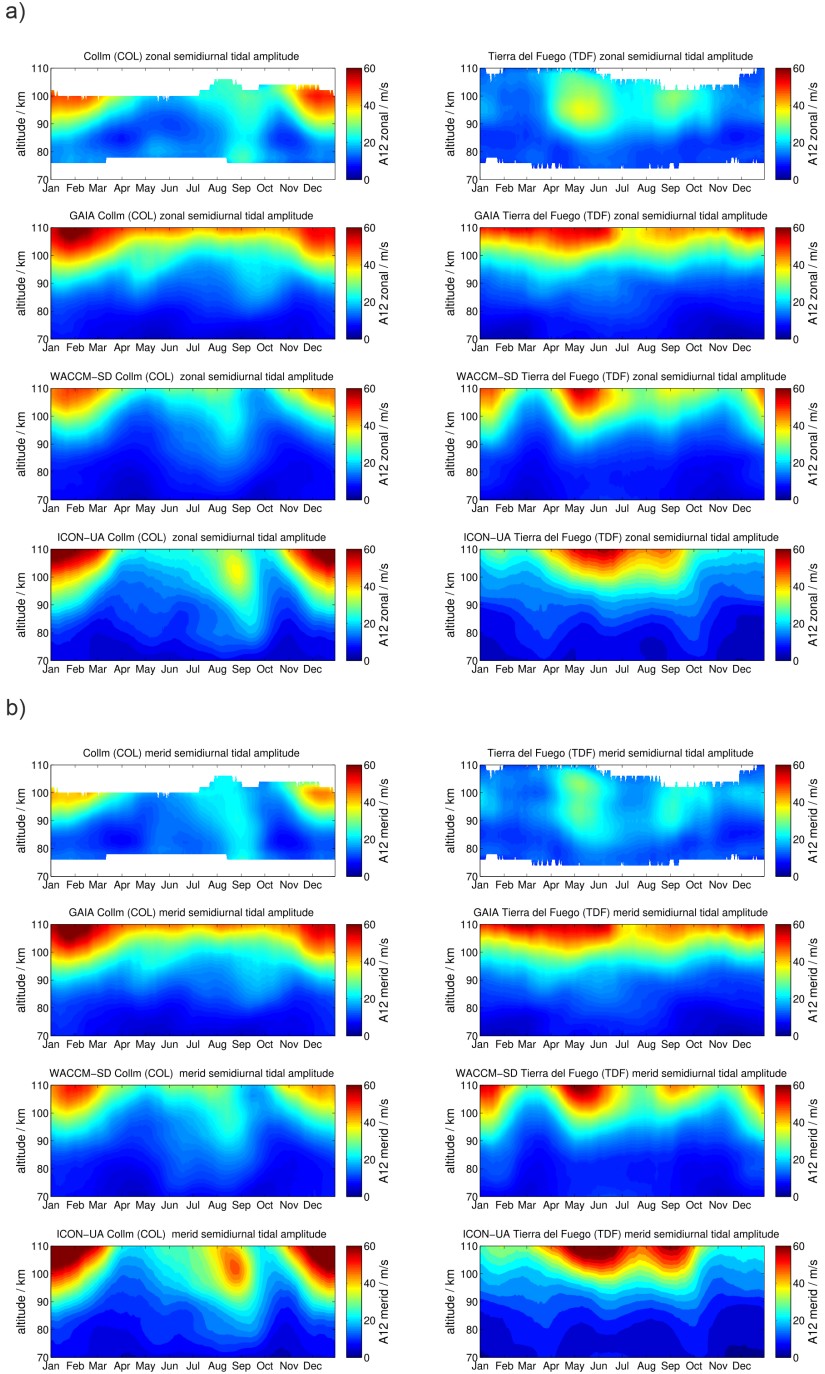

**Figure 13.** The same as Figure 2, but for the semidiurnal tidal amplitudes above COL and TDF.





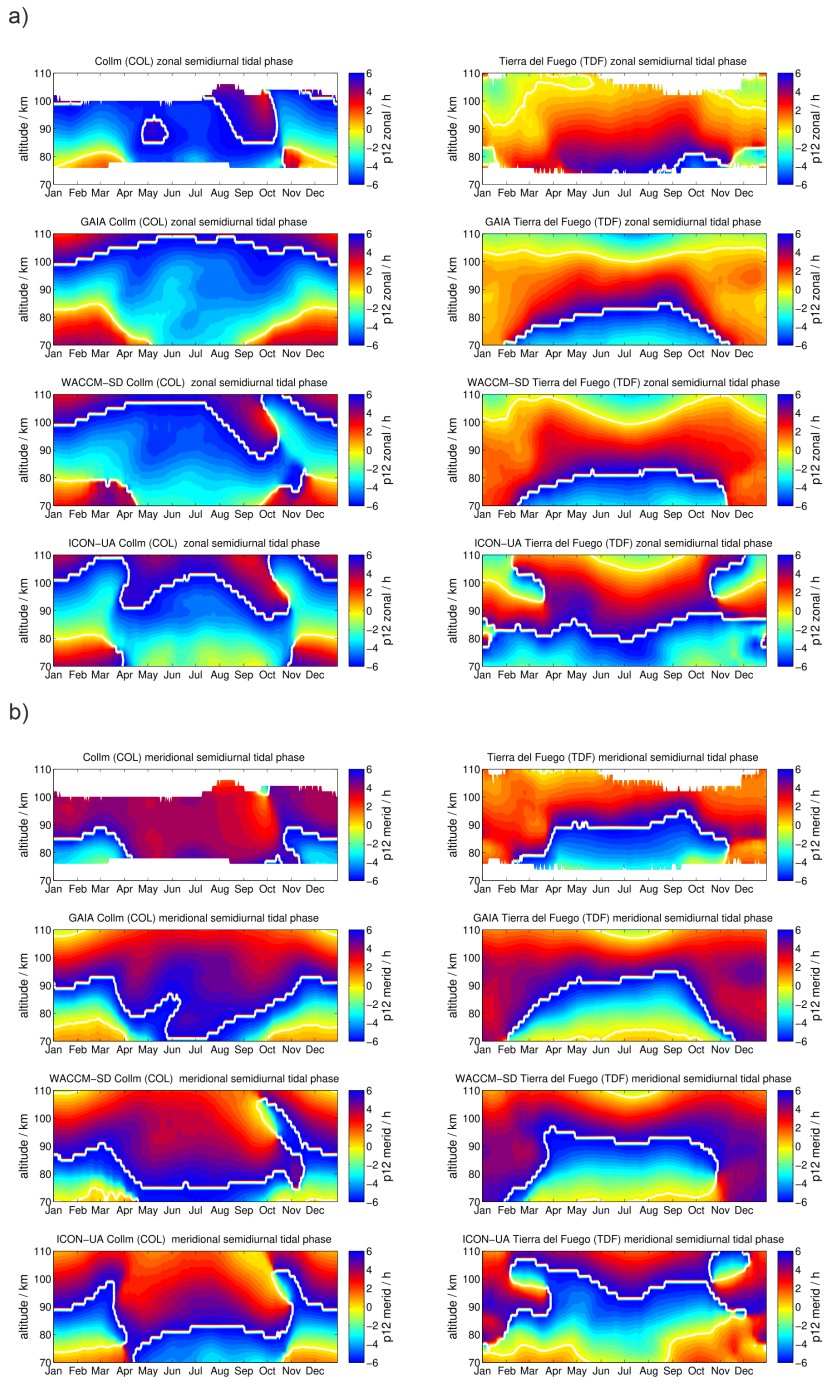

**Figure 14.** The same as Figure 2, but for the semidiurnal tidal phases above COL and TDF.





the much higher amplitudes during the winter months from November to February in the northern hemisphere. As expected
(e.g., Jacobi, 2012), the zonal and meridional amplitudes are almost the same. Apparently, TDF in the southern hemisphere
indicates a different seasonal amplitude characteristic. TDF exhibits still the largest amplitudes during the winter months from
April to October, but with much weaker amplitudes compared to COL. Furthermore, at TDF the zonal component shows larger
amplitudes than the meridional semidiurnal tide.

The GAIA model shows only a weak seasonality of the semidiurnal tidal amplitudes and even larger amplitudes above 100
km than the MR measurements. GAIA exhibits almost no interhemispheric differences of the tidal amplitudes. Only in the
northern hemisphere, GAIA indicates a semidiurnal tidal enhancement for the winter months, as it is found in the observations.
WACCM-X(SD) and UA-ICON indicate a rather good representation of the semidiurnal tide in the northern hemisphere for
the location of COL. The seasonality of the amplitudes are well-captured and exhibit a remarkably good agreement compared to
the observations. In the southern hemisphere the semidiurnal tides are less well represented in WACCM-X(SD). The model
shows a hemispheric summer tidal enhancement at altitudes above 90 km, which is missing in the observations. Furthermore,
the amplitudes appear to be increased relative to the observations at TDF. For the hemispheric winter months above TDF,
WACCM-X(SD) shows increased tidal amplitudes relative to the observations, but captures the general hemispheric winter
characteristic from May to July/August. Interestingly, UA-ICON indicates the best agreement with the observation for the sea-
sonality of the semidiurnal tidal amplitudes on both hemispheres and even reproduces the interhemispheric differences quite
well.

Semidiurnal tidal phases for the mid-latitude conjugate comparison are shown in Figure 14. The seasonal phase characteristic is
rather similar compared to the polar latitudes. The measurements as well as the models indicate a significant interhemispheric
difference that was already depicted in the amplitudes. On the northern hemisphere, we find a biannual seasonal phase charac-
teristic in the observations that is well-reproduced in the WACCM-X(SD) and UA-ICON data. GAIA also shows a reasonable
agreement, but does not reflect the quick phase change during the northern hemispheric fall transition in September/October.
In the southern hemisphere the observations at TDF show a more smooth seasonal phase characteristic that appears to be only
partially reproduced by the three models, which show distinguishable phase differences compared to the measurements.

Vertical wavelengths are shown in Figure 15. The observations indicate a clear seasonality of the vertical wavelengths
with much longer wavelengths during summer and during the fall and spring transitions. These long wavelengths seem to be
associated with the seasonal characteristic of the westward zonal winds. Our observations indicate that during the hemispheric
winter months vertical wavelengths of 40-90 km are common for the mid-latitudes. As expected from the phases, GAIA
shows the best agreement to the observations during the winter months and at the southern hemisphere. GAIA also shows the
long wavelengths in the northern hemisphere summer. WACCM-X(SD) and UA-ICON tend to show a reasonable agreement
with the northern hemisphere and, in particular, show the rapid phase change during the fall transition. However, UA-ICON and
WACCM-X(SD) also obtain similar wavelengths during the southern hemispheric winter months compared to the observations.





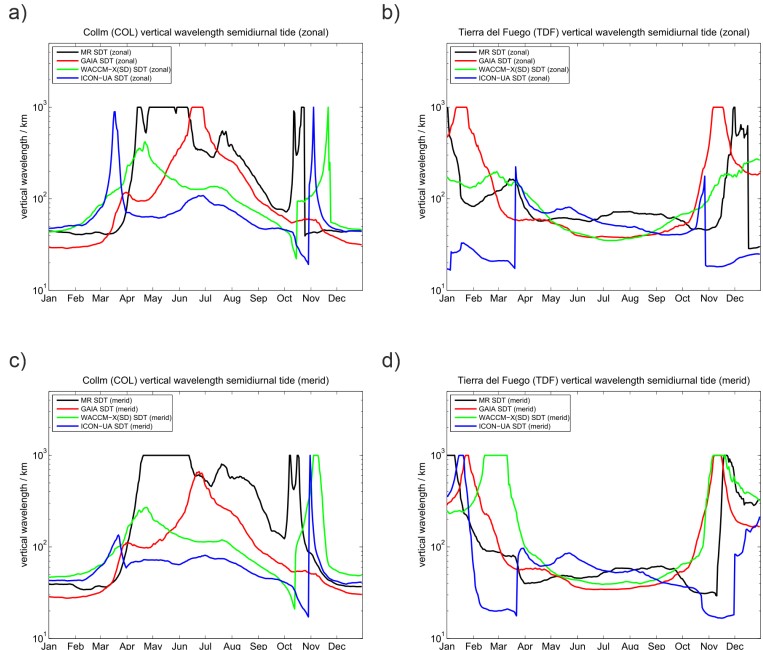

**Figure 15.** The same as Figure 8, but for COL and TDF.

## 6  Other stations

Finally, we investigate the KIR high-latitude meteor radar data and the CMOR observations in Canada, which is located at the lowest latitude used in this study. The KIR meteor radar is included as sanity check for the robustness of the meteor radar observations as it is located in the proximity of SOD. In addition, these two stations provide a good comparison to show how the seasonal characteristic of mean winds, diurnal and semidiurnal tides change with latitude. The data are displayed in Figures A1 - A7.

The mean wind and tidal climatologies at SOD and KIR are almost identical and, thus, show a similar agreement with GAIA, WACCM-X(SD) and UA-ICON. Comparing KIR and CMOR provides a more direct assessment of latitudinal differences. During the northern hemispheric winter CMOR observes an eastward zonal wind, which reaches higher magnitudes compared to KIR, but also indicates a wind reversal above approximately 100 km to a weak westward wind. The summer wind reversal from westward to eastward winds occurs at an almost 5-8 km lower altitude relative to KIR. GAIA, WACCM-X(SD) and UA-ICON reproduce the increased strength of the eastward winds during the winter season, but have difficulties to reproduce the summer zonal wind reversal. GAIA underestimates the eastward acceleration above 90 km, whereas WACCM-X(SD) and UA-ICON overestimate the eastward zonal wind during the summer. Meridional winds exhibit a clear annual characteristic above CMOR. The general seasonal characteristic can be also found in GAIA, WACCM-X(SD) and UA-ICON.

However, the diurnal tidal amplitudes and phases appear to be more difficult to be captured by the models for the CMOR location compared to the MR observations. WACCM-X(SD) and UA-ICON indicate a better agreement of the diurnal tidal





seasonal amplitude behaviour relative to GAIA, but both models exhibit larger differences in the diurnal tidal phases above 90 km concerning the CMOR observation. This is also the case for the semidiurnal tidal amplitudes and phases. Both models indicate a less good agreement with the observations of the seasonal characteristic of the semidiurnal tidal amplitude. However,

the phases of the semidiurnal tide indicate a better agreement with the observations. In particular, WACCM-X(SD) captures the seasonal variability during the fall transition remarkably well. UA-ICON tends to show larger dissimilarities compared to the observations and the WACCM-X(SD) for the CMOR locations.

## 7 Discussion

The mean circulation at the MLT is substantially controlled by tides and gravity waves, carrying energy and momentum from

their source regions to the altitude of their dissipation (e.g., Lindzen, 1981a, and references therein). Gravity wave dissipation drives the hemispheric summer mesopause temperatures up to 100 K away from the radiative equilibrium (McLandress et al., 2006; Becker, 2012; Sato et al., 2018). However, GCMs often do not have the spatial and temporal resolution to resolve GWs and, thus, depend on GW parameterizations of the different primary GW sources such as orography (mountain waves), frontal systems, jet stream imbalances, deep convection and shear instabilities (Fritts and Alexander, 2003; Plougonven and Zhang,

2014, also see for a review). Recently, there were studies suggesting that non-primary GWs also contribute to the momentum budget at the MLT (Vadas and Becker, 2018; Vadas et al., 2018; Becker and Vadas, 2018; Sato et al., 2018) resulting in an even more complex vertical coupling posing new challenges not yet considered in available GW parameterizations.

GAIA and WACCM-X(SD) are both incorporating similar GW parameterizations for orographic and non-orographic primary GW based on (McFarlane, 1987b; Lindzen, 1981b). Although the applied schemes are supposed to be comparable, the re-

sulting forcing is rather dissimilar concerning the magnitude of the parameterized GW drag. UA-ICON uses different GW parameterizations (Giorgetta et al., 2018; Hines, 1997; Lott, 1999) which are, however, structurally similar with typical simplifying assumptions like pure vertical and instantaneous propagation of GWs. Pedatella et al. (2014a) estimated the mean zonal GW drag from GAIA and WACCM-X applying the relation shown in Liu et al. (2009b). The net zonal mean GW drag was more than twice as large in WACCM-X(SD) compared to GAIA at mid- and polar latitudes from the stratosphere up to the

mesosphere. The comparison also indicated differences in the vertical structure of where the GW drag accelerates/decelerates the mean zonal wind and, hence, explains why the mean wind structures are different between GAIA and WACCM-X(SD). As the GW parameterization of UA-ICON is similar to the HAMMONIA model which has been shown by (Pedatella et al., 2014a) to produce mesospheric GW drag smaller than WACCM-X but larger than GAIA. Apparently, the parameterization in WACCM-X(SD) seems to be more suitable to reproduce the hemispheric summer mesopause zonal wind reversal, whereas the

GAIA GW parameterization is more adequate for hemispheric winter conditions.

In the past, GCMs were often validated and cross-compared to other global data sets such as satellite observations using typical winter or summer conditions and comparing zonal means as vertical-latitude cross-sections (McLandress et al., 1996; McLandress, 1997; Du et al., 2007; Liu et al., 2009b; Smith, 2012; Liu et al., 2013; Pedatella et al., 2014a). Ground-based observations open the possibility to perform a better climatological comparison, but only for a given location. McCormack





et al. (2017) conducted a cross-validation between the meteorological analysis with NAVGEM-HA and globally distributed MR wind measurements using altitude vs. time plots to access the temporal and short term variability of mean winds and tides for two winter seasons. Later, Stober et al. (2019) performed a cross-comparison of the seasonal climatology for three MR stations and a more detailed verification of the tidal variability of the tidal amplitude and phase behaviour using NAVGEM-HA and MRs. These comparisons revealed a remarkable agreement between the observations and the meteorological analysis on interday to seasonal time scales for mean winds as well as for atmospheric tides providing confidence in both data sets to capture realistic atmospheric dynamics in the MLT.

In this study, we present the first systematic investigation of interhemispheric differences of mean winds and atmospheric tides at conjugate latitudes from observations and comprehensive models applying a unified diagnostic. Although the results are in line with other studies using MR observations to evaluate mean winds and tides in the MLT, they reveal in more detail the latitudinal differences between both hemispheres and present a more systematic evaluation of state of the art models at the MLT. Pancheva et al. (2020) compared MR observations from Svalbard and Tromsø to WACCM-X and CMAM-DAS and found similar agreements and deviations concerning the seasonality of mean winds and tides. In particular, their results obtained from Tromsø are comparable to SOD and KIR. Sato et al. (2018) compared GAIA to Microwave Limb Sounder (MLS) temperatures and geostrophic winds for January conditions and found a very good agreement from 12 km altitude up to 80 km for the southern hemisphere, and up to 97 km for the northern hemisphere. This comparison also indicated a too weak and high up summer zonal wind reversal for the southern latitudes in GAIA in contrasts to Yasui et al. (2018) stating that the missing wind reversal may be due to the limitation of the representation of the parameterized gravity waves in the model. Some of the differences between GAIA, WACCM-X(SD) and the MR observations have their origin further down in the atmosphere. Both models are nudged to the reanalysis data. GAIA is nudged to the JRA-25 and from 2015 on to JRA-55 reanalysis data (Kobayashi et al., 2015) up to 30 km, whereas WACCM-X(SD) is driven by MERRA (Rienecker et al., 2011) up to 50 km. Harada et al. (2016) performed a cross-comparison among various reanalysis data sets to investigate potential differences and found that JRA-25, JRA-55 and MERRA already showed some differences in storm tracks and mean winds and temperatures. In particular, at the upper stratosphere the reanalysis data sets and observation from MLS indicated some differences among each other. Due to the nudging of GAIA and WACCM-X(SD), these differences enter also into the model fields and partly explain differences among the models as well as concerning the MR observations. The nudging height may also play some role since the systematic differences have been found in the upper stratosphere (Sakazaki et al., 2018).

Furthermore, the reanalysis data used for the nudging of GAIA and WACCM-X(SD) seems to be less relevant for the representation of the diurnal and semidiurnal migrating tide climatologies. Ortland (2017) estimated the tidal forcing for the DW1 and SW2 due to absorption of solar radiation from ozone and water vapor. They found that the tidal correspondence between the Tide Mean Assimilation Model (TMAT) and observations at the MLT strongly depend on the forcing at the troposphere and stratosphere for both migrating tides. Thus, differences in the tides between the models are most likely the result of different implementations of the radiative transfer and distribution of tropospheric/stratospheric water vapor and ozone causing differences in the radiative forcing. This is further supported by the free running UA-ICON, which shows very good agreement to the tides produced in WACCM-X(SD), but is not driven by any reanalysis data.





Another aspect worth to be discussed are the tidal phases. Apparently, WACCM-X(SD), UA-ICON and GAIA reach a fairly good agreement with the semidiurnal tidal phases at mid- and polar latitudes capturing many of the seasonal characteristics that are found in the MR observations. It is also obvious that GAIA tends to better agree on the southern hemisphere and WACCM-X(SD) performs a bit better on the northern hemisphere. Only the diurnal tidal phases at polar latitudes are dissimilar in both models compared to the MR observations. The vertical wavelength of the diurnal tides in GAIA and WACCM-X(SD) suggest

almost evanescent diurnal tidal modes, whereas the observations indicate much shorter wavelengths and a vertically propagating diurnal tide. Both models reduce the longitudinal grid resolution closer to the pole to avoid the singularity. This seems to favour a damped vertical propagation of the diurnal tide, but has to be investigated in more detail.

Non-migrating tides are also worth to discuss. In fact, our MR observations are local and, thus, the observed diurnal and semidiurnal tides are a superposition of the migrating and non-migrating tidal modes. At the lower latitudes, the generation of

non-migrating tides is well-understood due to the latent heat release in the tropics (Hagan and Forbes, 2002, 2003; Oberheide et al., 2011). At mid- and polar latitudes non-migrating tides can be generated by various processes such as latent heat release, nonlinear interactions with stationary planetary waves (Yamashita et al., 2002; Smith et al., 2007; Murphy et al., 2009; Miyoshi et al., 2017) and other tidal modes, variations in the mean background wind and temperature field, and gravity wave breaking or dissipation regions (Fritts et al., 2006). Furthermore, there were some studies investigating SSW as a potential cause to excite

the westward propagating semidiurnal tides with wavenumbers 1 and 3 (Du et al., 2007; Liu et al., 2010b; Stober et al., 2020a). There are only a few studies investigating the climatology of non-migrating tides available using ground-based networks. Murphy et al. (2006) presented a climatoglogy of the diurnal and semidiurnal tides for the southern hemisphere combining several radars located at Antarctica. Their results indicate that the amplitudes of the non-migrating components are between 1-5 m/s and occasionally reach up to 10 m/s. However, this study also confirmed that the migrating component is most of the time

of the year the dominating tide, which was also found in Baumgarten and Stober (2019), during a campaign on the northern hemisphere. Recently, there were some studies on the northern hemisphere based on SuperDARN observations to derive climatologies of the migrating (van Caspel et al., 2020) and non-migrating tides (Hibbins et al., 2019). These studies revealed amplitudes of about 5 m/s throughout the year for the non-migrating components and amplitudes of about 15-20 m/s for the migrating tidal components at high polar latitudes. Considering the various non-migrating tidal generation mechanism it is ob-

vious that there is only a weak coherence between the years due to phase variability of their source processes (e.g. SSWs occur at different dates from year to year at the northern hemisphere). Hence, the obtained tidal climatological comparison presented here, is dominated by the migrating tidal components. The comparison of the free-running UA-ICON model with the nudged GAIA and WACCM-X(SD) indicates that the climatology of mean winds and tides at the MLT are driven by the model physics and do not strongly depend on the nudging at the troposphere/stratosphere. Apparently UA-ICON and WACCM-X(SD) employ

GW parameterizations that yield a similar climatological wind at the MLT, although the detailed implementation is different. Furthermore, the good agreement of the tidal amplitude and phases between WACCM-X(SD) and UA-ICON suggests that the seasonal characteristics and the resultant tidal climatologies are also less dependent on the nudging. Recently, Dempsey et al. (2021) performed a cross-comparison of mean winds and tides in the southern hemisphere using the meteor radar at Rothera and WACCM as well as the Extended Canadian Middle Atmosphere Model (eCMAM). They postulated that non-primary





waves play a role for the MLT winds. Theoretical studies with the Kuehlungsborn Mechanistic Circulation Model (KMCM) showed that secondary or non-primary wave generation provides an essential contribution to the MLT wind forcing above GW hot spots such as the southern Andes and the Antarctic Peninsula (Becker and Vadas, 2018; Vadas and Becker, 2018). A first observational evidence was obtained at McMurdo investigating lidar observations (Vadas et al., 2018). More recently a detailed study of the MLT dynamics for the year 2019 using six meteor radars from Tierra del Fuego, South Georgia, Rothera, King

Sejong Station, Davis and McMurdo indicated a strong impact of non-primary waves above the Andes and Antarctic Peninsula on the daily mean zonal and meridional winds and momentum fluxes (Stober et al., 2021). Including non-primary waves into the GW parameterizations for the hemispheric winter could add eastward momentum to the MLT, which is apparently too weak in WACCM-X(SD) and UA-ICON. The strong winter stratospheric eastward polar vortex efficiently removes all eastward GW by critical level filtering and, thus, only westward GW propagate to the mesosphere and can deposit their momentum resulting

in an westward forcing and westward mean winds. Considering non-primary waves in the parameterization could essentially balance the total forcing at the MLT and may help to get the mean winds to a better agreement with the observations.

## 8    Conclusions

In this study we compared GAIA, UA-ICON and WACCM-X(SD) predictions with local meteor radar observations applying a

unified diagnostic to decompose the wind field into mean winds, as well as diurnal and semidiurnal tidal amplitudes and phases in the MLT. Therefor we present observations from six meteor radars and derived climatologies from the continuous observations for the above mentioned meteorological parameters, which are cross-compared to nudged model simulations from GAIA and WACCM-X(SD) for the same periods as the measurements are available from each radar. In addition, a 21 year UA-ICON free-running GCM run was employed for comparison.

Although all models utilize similar gravity wave parameterizations schemes, but different implementations, the zonal and meridional winds exhibit seasonal and interhemispheric differences between GAIA, WACCM-X(SD), UA-ICON and the MR observations. It is obvious that GAIA shows a better agreement of mean winds during the winter season in both hemispheres compared to the meteor radar, whereas WACCM-X(SD) and UA-ICON indicates a better agreement of the zonal wind reversal from westward to eastward in both hemispheres with the observations. However, one has to note that GAIA seems to produce a

too weak and high up hemispheric summer zonal wind reversal. WACCM-X(SD) and UA-ICON tend to show westward winds during the winter season pointing towards too much westward wave drag at these altitudes. Furthermore, meridional winds appear to be in remarkable agreement between GAIA and the MR observations at mid- and polar latitudes, while WACCM-X(SD) and UA-ICON indicates more dissimilarities in the meridional winds relative to the MR observations. UA-ICON, as a free-running model, nevertheless shows remarkably good representation of MLT wind fields compared to the two nudged

models.

The ASF decomposition of the time series from the models and the meteor radar observations ensures a harmonized tidal comparison of the amplitude and phases. Atmospheric tides provide an essential source of variability for the coupling of the



middle atmosphere to the ionosphere. Daily tidal amplitudes and phase are obtained from the ASF and vector averaged, which reduces the contamination of the amplitude and phase due to the tidal intermittency caused by non-linear wave-wave interac-

tions, changes in the mean winds or source variabilities. There is a good agreement of the GCMs for the diurnal tide amplitude and phase for the latitudes investigated here in. Diurnal tides indicate only weak interhemispheric differences and reach the largest amplitudes above 95-100 km during the hemispheric summer months. The seasonality of the diurnal tidal amplitudes is well-reproduced by GAIA, UA-ICON and WACCM-X(SD). However, diurnal tidal phases show some differences between the observations and the GCMs. GAIA, UA-ICON and WACCM-X(SD) tend to exhibit much longer vertical wavelengths

compared to the MR measurements.

Semidiurnal tides are the dominating tidal mode at mid- and high-latitudes through the course of the year and MLT altitudes. One of the main results of this comparison are the distinct differences of this tide between both hemispheres. It appears that for conjugate latitudes the semidiurnal tide reaches higher amplitudes in the northern hemisphere at mid- and polar-latitudes. In particular, the amplitude and enhancement and phase variability in September on the northern hemisphere is not found at the

southern latitudes during the transition from the hemispheric summer to the winter circulation. More detailed investigations are required to distinguish potential reasons, which are likely caused by a complex chain of interactions due to the differences in the land-sea distribution, GW sources and planetary waves between both hemispheres that alter the polar vortices and, thus, the ozone transport into the polar cap, which again provides a feedback on the excitation of tides. GAIA, UA-ICON and WACCM-X(SD) indicate a reasonable agreement of the semidiurnal tidal amplitude and phase. There is a tendency that WACCM-X(SD)

and UA-ICON has a better agreement with the MR observations on the northern hemisphere, whereas GAIA seems to agree better on the southern hemisphere. However, all GCMs have a tendency to overestimate the summer hemisphere semidiurnal tidal amplitudes above 100 km.

The climatological comparisons of mean winds and diurnal and semidiurnal tides underline the value of continuous observations in the MLT to evaluate/assess GCMs. GAIA, WACCM-X(SD), and UA-ICON are state of the art models, coupling the

middle atmosphere with the upper atmosphere to study the forcing from below of the thermosphere/ionosphere system and a potential feedback to the middle atmosphere. Therefore, we assessed the climatological state of the mean winds and the tidal activity at the MLT. We identified systematic dissimilarities in the mean zonal and meridional winds and in the seasonal characteristic of tidal amplitudes and phases. However, there was a remarkable agreement in both hemispheres of the semidiurnal tide between the observations and the free-running UA-ICON, which further underlines that the climatological behavior at the

MLT seems to be not driven/improved by the nudging of GCMs to reanalysis data.





## Appendix A: Mean winds

Appendix A1, A2, etc.

*Author contributions.* The conceptual idea of the manuscript was developed by GS, DP, CJ and HuL. The data analysis and data reduction was performed by AKu and DP. HuL supported the data analysis and interpretation of GAIA and contributed to the discussion. HLi supported the interpretation of the WACCM-X(SD) results and the overall discussion. HS provided the UA-ICON data and helped with the interpretation of the results. KB provided an essential contribution developing the ASF method. All authors contributed to the editing and writing of the manuscript. AK and ML provided SGO meteor radar data. EB and JK contributed the Esrange meteor radar observations. DJ shared the TDF meteor radar measurements and DM supported this work by providing DAV data. PB provided the CMOR radar data.

*Competing interests.* The authors declare that there are no competing interests.





a)



b)

**Figure A1.** The same as Figure 2, but for the mid-latitudes at KIR and CMO.





a)



b)

**Figure A2.** The same as Figure 2, but for the diurnal tidal amplitudes above KIR and CMO.





a)

b)

**Figure A3.** The same as Figure 2, but for the diurnal tidal phases above KIR and CMO.





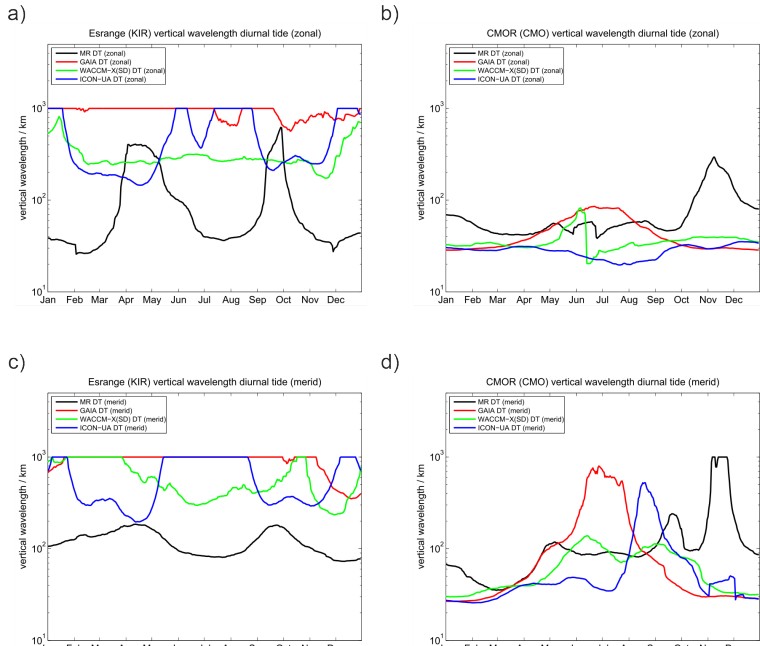

**Figure A4.** The same as Figure 5, but for KIR and CMO.



a)

b)

**Figure A5.** The same as Figure 2, but for the semidiurnal tidal amplitudes above KIR and CMO.





a)

b)

**Figure A6.** The same as Figure 2, but for the semidiurnal tidal phases above KIR and CMO.



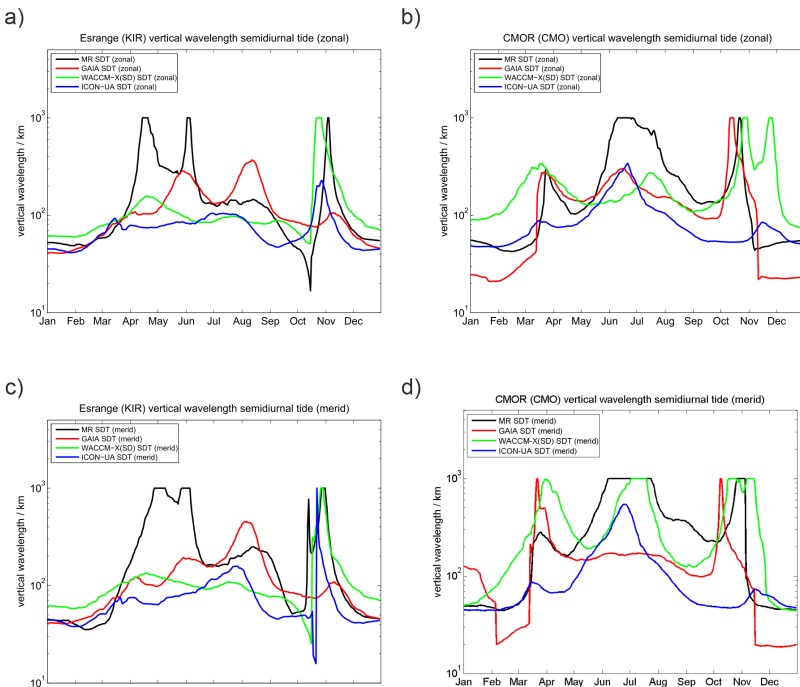

**Figure A7.** The same as Figure 8, but for KIR and CMO.



*Acknowledgements.* Gunter Stober is a member of the Oeschger Center for Climate Change Research (OCCR). The Esrange meteor radar operation, maintenance and data collection is provided by Esrange Space Center of Swedish Space Corporation. AK and CJ acknowledge support by Deutsche Forschungsgemeinschaft through grant JA 836/43-1. HuL acknowledges supports by JSPS KAKENHI grants 18H01270, 18H04446, 17KK0095, and JRPs-LEAD with DFG. Operation of the Davis Meteor radar is supported through Australian Antarctic Science projects 2668, 4025 and 4445. Diego Janches was supported by the NASA Heliophysics ISFM program. TDF's operation is supported by

NASA SSO, NESC assessment TI-17-01204, and NSF grant AGS1647354. This work was supported in part by the NASA Meteoroid Environment Office under cooperative agreement 80NSSC18M0046. PGB also acknowledges funding support from the Natural Sciences and Engineering Research council of Canada (RGPIN- 2016-04433) and the Canada Research Chairs program (grant 950-231930).





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
