# Peer review of "Interhemispheric differences of mesosphere/lower thermosphere winds and tides investigated from three whole atmosphere models and meteor radar observations"

_Atmospheric Chemistry and Physics, 2021_

## Author Response (AR1)

Reply to reviewer #1

We thank the reviewer for the comments on our manuscript. The revised manuscript has now one more co-author to acknowledge the support of the Esrange meteor radar. We added additional figures to the appendix indicating absolute differences between the meteor radar and the three models for daily mean winds and tides.

**General Comment:**

This study compares model simulations with meteor radar (MR) observations. it presents the first systematic investigation of interhemispheric differences of mean winds and atmospheric tides at conjugate latitudes from observations and comprehensive models applying a unified diagnostic. It is a much-needed study. While I appreciate the heroic effort of the authors, I am troubled by the lack of similarity between the model simulations described here and the MR observations. Previous studies - as properly discussed in this manuscript - revealed a much closer match between MR data and NAVGEM-HA. I am troubled by how badly the models are doing here.

**General reply:**

The disagreement between the observations and nudged and free-running models at the MLT is obvious. The main difference between NAVGEM-HA and the nudged models is the data assimilation and the assimilated data sets. NAVGEM-HA obtains meteorological fields using a 4DVAR scheme and observations up to the model top, which is by far more complicated than the typical nudging to reanalysis data for WACCM-X(SD) and GAIA up to a pre-defined altitude at the stratosphere-stratopause. It is obvious that the model climatologies at the MLT are not affected by the nudging and, thus, the climatologies of meteorological fields represent more the free-running state of the GCM models.

**Comment:**

The authors discuss the different behavior of the gravity wave drag parameterization in these models, arguing that the mean circulation is different to partially explain the differences, I believe. I don't disagree in general, however, such discrepancies seem to point to a fundamental flaw in these models: the lack of observations at MLT altitudes. Isn't that the take-home message of this study?

**Reply:**

We think there are two aspects here that are important. It is certainly true that a denser network of MLT observations would improve the quality of the meteorological fields, when data assimilated and included in NAVGEM-HA using a 4DVAR scheme. The other aspect that is worth mentioning, but much more difficult to capture, are the reasons why the models apparently generate these dissimilarities. GW parameterizations are playing a key role here. There is a lack of process understanding at altitudes between 50-90 km about the gravity wave dynamics that need to be resolved and considered in the parameterization schemes. More and better observations are required to address the missing physics e.g., Voelker et al., 2021 (Voelker, GS, Akylas, TR, Achatz, U. An application of WKBJ theory for triad interactions of internal gravity waves in varying background flows. *Q J R Meteorol Soc*. 2021; 147: 1112– 1134. https://doi.org/10.1002/qj.3962).

**Comment:**

Moreover, some of the models use atmospheric specifications (like MERRA): what is the time cadence of these atmospheric specifications? Typically these data are provided 6-hourly, which would not resolve semidiurnal variability, and in such case, the comparison is between observations and the model's own climatology. In the same spirit, what it the nudging time scale? A long-timescale would prevent the model to be tightly associated with the atmospheric analysis.

**Reply:**

WACCM-X(SD) performs the nudging every 3 hours using MERRA2 observations. The model states are nudged to the target states with a time scale of 50 hours (up to 50 km). The nudging procedure is described in Smith et al (2017, 10.1175/JAS-D-16-0226.1). We added a sentence referring to nudging in the WACCM-X(SD) section.

Specific comments:

**Comment:**

Page 3, bottom: Why McCormack et al. (2015). That's a QBO paper. I think you want to use McCormack et al. (2017) as in the rest of the manuscript.

**Reply:**

Corrected.

**Comment:**

Figure 3 and similar figures. The authors really need to add contours: the color palette has a very large dynamic range and for the reader it is impossible to discern contours, especially when the largest values are at the edges of the panels and the vast majority of the figure is a bland uniform color. Also, I think would it help explaining the figures (and for us the readers, understanding it) if one hemisphere (say the SH) is rotated by 6 months, so that the same season is always in the middle.

**Reply:**

All figures were revised according to the suggestion. We added labelled contour lines for the tidal amplitudes and shifted the time axis by 6 months for the southern hemispheric stations.

**Comment:**

Page 10, middle. Why is it expected that conjugate latitudes see almost the same behavior?

**Reply:**

We rephrased this sentence and explained that a certain level of agreement between both hemispheres is expected due to the residual circulation.

Reply to reviewer #2

We thank the reviewer for the comments on our manuscript. The revised manuscript has now one more co-author to acknowledge the support of the Esrange meteor radar. We added additional figures to the appendix indicating absolute differences between the meteor radar and the three models for daily mean winds and tides.

**General Comment:**

The paper is a solid and well-founded comparison between northern and southern hemispheric MLT observations on the one hand and the comparison of the model output of three whole atmosphere GMCs with the observations on the other hand. The structure of the paper is logical and clear, but resembles more a technical report than a scientific paper.

The main presentation of the results is a mixture of contrasting observations from the two hemispheres for different locations and the same presentation for the model results (10 of 15 figures do this plus the five in the Appendix). As the technical methodology appears to be very sound, I only have two main remarks that might help modify the current manuscript and improve the presentation:

**General reply:**

This paper is prepared as part of the VACILT project. The main goal is to document the climatological state of the harmonized meteor radar time series and corresponding continuous nudged and free running GCM-models. We plan to analyze the compiled time-series concerning long-term changes and certain meteorological events in additional publications. We agree with the reviewer statement that the manuscript resembles more a technical report about the wind and tidal climatologies, which may be helpful for model developers to optimize the GW parameterizations to reduce dissimilarities between the models and the observations.

**Comment:**
(a) I would suggest separating the physical comparison between the hemispherical observations from the comparison GCMs to the observations. This would allow the authors to formulate research questions that can be addressed and answered by the comparison. The comparison between the GCMs and the observations should constitute a second main part of the paper. Currently, it is a hodgepodge, hard to read and difficult to separate the individual results.

**Reply:**

We would like to keep the structure of the manuscript as it is, mainly to foster the model development of GW parameterizations. We intentionally did not add more scientific information and data interpretation as it would further reduce the readability. However, we changed the time axis for all plots of the southern hemispheric stations to permit an easier comparison of the zonal winds and tidal climatologies concerning the seasons.

**Comment:**

(b) The authors go to great lengths to create a homogeneous data set consisting of both observations and model outputs on comparable altitude-time grids. I wonder why the results are presented and discussed only qualitatively (" ... agrees reasonably well ...", etc). Why don't the authors show differences of the climatological means MODEL vs. OBSERVATION? I admit that the authors use a lot of effort to turn the shocking disagreements into positive words (e.g. " ....  shows a better agreement with the radars for the hemispheric zonal summer wind reversal ..." ) but for scientific usage a QUANTIFICATION of the differences would be really desirable!

**Reply:**

We added qualitative differences of mean winds, and tidal amplitudes for all stations in the appendix. These images document already the dissimilarities, therefore we did not intend to put more emphasis on that by our wording. However, we point out that an agreement of the seasonal morphology is more important for comprehensive GCMs concerning the physics. Small altitude differences can already lead to large absolute differences, although the underlying process might be correct and just the parameterizations need to be optimized.

**Comment:**

Nevertheless, the study has its merits but really needs focus.

**Reply:**

We revised the conclusions to put more emphasis on the main results.

---

## Author Response (AR2)

Authors reply:

We thank both reviewers for the helpful and constructive comments on our publication and that they recommended the manuscript after the revision as ' publish as is'.